# Reversal of pre-existing NGFR-driven tumor and immune therapy resistance

Julia Boshuizen [1], David W. Vredevoogd [1,7], Oscar Krijgsman[1,7], Maarten A. Ligtenberg[1], Stephanie Blankenstein[2], Beaunelle de Bruijn[1], Dennie T. Frederick[3], Juliana C. N. Kenski[1], Mara Parren [1], Marieke Brüggemann[1], Max F. Madu[2], Elisa A. Rozeman[1], Ji-Ying Song[4], Hugo M. Horlings [5], Christian U. Blank [1], Alexander C. J. van Akkooi[2], Keith T. Flaherty [6], Genevieve M. Boland [3] & Daniel S. Peeper [1]✉

Melanomas can switch to a dedifferentiated cell state upon exposure to cytotoxic T cells. However, it is unclear whether such tumor cells pre-exist in patients and whether they can be resensitized to immunotherapy. Here, we chronically expose (patient-derived) melanoma cell lines to differentiation antigen-specific cytotoxic T cells and observe strong enrichment of a pre-existing NGFR[hi] population. These fractions are refractory also to T cells recognizing non-differentiation antigens, as well as to BRAF + MEK inhibitors. NGFR[hi] cells induce the neurotrophic factor BDNF, which contributes to T cell resistance, as does NGFR. In melanoma patients, a tumor-intrinsic NGFR signature predicts anti-PD-1 therapy resistance, and NGFR[hi] tumor fractions are associated with immune exclusion. Lastly, pharmacologic NGFR inhibition restores tumor sensitivity to T cell attack in vitro and in melanoma xenografts. These findings demonstrate the existence of a stable and pre-existing NGFR[hi] multitherapy-refractory melanoma subpopulation, which ought to be eliminated to revert intrinsic resistance to immunotherapeutic intervention.

[1] Division of Molecular Oncology and Immunology, Oncode Institute, The Netherlands Cancer Institute, Amsterdam, The Netherlands. [2] Division of Surgical Oncology, The Netherlands Cancer Institute, Amsterdam, The Netherlands. [3] Department of Surgical Oncology, Massachusetts General Hospital, Boston, MA, USA. [4] Division of Animal Pathology, The Netherlands Cancer Institute, Amsterdam, The Netherlands. [5] Division of Pathology, The Netherlands Cancer Institute, Amsterdam, The Netherlands. [6] Department of Medical Oncology, Massachusetts General Hospital, Boston, MA, USA. [7] These authors contributed equally: David W. Vredevoogd, Oscar Krijgsman. ✉email: d.peeper@nki.nl

The landscape of treatment regimens for late stage melanoma patients has shifted progressively in recent years, largely owing to the development of therapies designed to generate or enhance T cell-mediated tumor killing. Clinical benefit of immune checkpoint blocking antibodies, such as anti-PD-1 and anti-CTLA4, has been reported to be over 50% in melanoma[1–3]. Also other T cell-based therapeutic modalities, such as adoptive T cell transfer (ACT) of either endogenous or genetically engineered T cells, have shown durable responses in a subset of melanoma patients[4–8]. However, the majority of tumors display either innate or acquired resistance to these therapies, due to highly pleiotropic mechanisms including lack of actionable and clonal antigens and tumor heterogeneity[9–11]. Currently, there is still no full understanding of how these mechanisms contribute to immunotherapy resistance, especially in the context of intratumor heterogeneity.

This is of particular importance in melanoma, a highly heterogeneous cancer type exemplified not only by the frequent presence of subclonal genetic alterations, but also by intratumoral transcriptional differences between melanoma cells, corresponding to different cell states[12,13]. We and others previously described one such cell state, which is characterized by high expression of the receptor tyrosine kinase AXL and low expression of the master melanocyte transcription factor microphtalmia-associated transcription factor and its downstream target MART-1/Melan-A[14,15]. Functionally, this process of dedifferentiation or phenotype switching is associated with both enhanced tumor invasion and resistance to MAPK pathway inhibitors[14–17]. We found that AXL was commonly expressed in heterogeneous patterns in clinical tumor samples, indicating that cell states in melanoma are also highly heterogeneous in patients[18].

Phenotype switching and dedifferentiation have been linked also to acquired T cell resistance, given that microenvironment-derived cytokines such as tumor necrosis factor (TNF) can cause reversible downregulation of melanocytic differentiation antigens and resistance to cognate T cells[19–21]. This plasticity is seen both in animal models and melanoma patients and is associated with expression of Nerve Growth Factor Receptor (NGFR), a protein originally identified as a putative melanoma stem cell marker[22,23]. NGFR has also been suggested to be a key regulator of phenotype switching[24]. Recent reports indicate that cell state heterogeneity in melanoma may be even more complex, with at least four dynamic cell states that can follow reversible trajectories[13,25].

In spite of these advances, it is currently unknown whether resistant melanoma cells that emerge upon chronic exposure to cytotoxic T cells pre-exist in patients prior to treatment, which we examined here. We also investigated whether such T cell-resistant melanoma cells display broader therapy resistance and whether they could be resensitized to T cells in vitro and in vivo. Lastly, we examined any association between the T cell-resistant cell state and the response to immunotherapy in melanoma patients. We describe a pre-existing cell population characterized by high expression of NGFR, which displays innate resistance to cytotoxic T cells as well as their cytokines. Moreover, NGFR[hi] cells display cross-resistance to a variety of other therapies, including BRAF and MEK inhibition, in vitro, in mice and in patients. These melanoma cells express high levels of the neurotrophic factor brain-derived neurotrophic factor (BDNF), and inhibition of either BDNF or NGFR enhances sensitivity to T cell-mediated tumor killing. Conversely, elevating the levels of NGFR in NGFR[lo] cells confers T cell resistance. Lastly, pharmacological inhibition of NGFR restores T cell sensitivity in tumor cells. These findings considerably extend our understanding of the importance of distinct melanoma cell states, in particular for NGFR[hi] cell fractions, in governing immune resistance. Our results may warrant further exploration in a clinical, therapeutic setting.

## Results

**Melanoma fractions resistant to antigen-specific T cells**. Since immunotherapy resistance is an increasing therapeutic challenge in cancer, including melanoma, we set up an experimental system to model T cell resistance, similar to what we and others have done successfully previously to study BRAF + MEK resistance[26]. We engineered a collection of matched human melanoma : T cell pairs to establish tumor cell populations that spontaneously acquired T cell resistance. Specifically, we subjected a series of established and patient-derived xenograft (PDX)-derived melanoma cell lines[27], which all endogenously express HLA-A*02:01 and MART-1, in vitro to healthy donor CD8 T cells that had been retrovirally transduced with a T cell receptor recognizing MART-1[28]. These melanoma cell lines initially displayed high sensitivity to the T cells, being largely eliminated in a 24 h co-culture (Fig. 1a, b, Supplementary Fig. 1a). However, a small fraction of tumor cells remained viable after this treatment. We expanded these rare cells and subsequently repeated the cycles of T cell challenge up to 15 times. This stringent selection procedure yielded six independent melanoma populations from as many different parental cell lines, which had spontaneously acquired resistance to T cells ("TR" cell lines; Fig. 1a, b, Supplementary Fig. 1a). The observed resistance phenotype could be recapitulated in mice: whereas tumors comprising parental melanoma cells responded readily to adoptive transfer of MART-1 T cells and underwent regression, melanomas produced by the derivative TR cells were fully resistant to T cell attack in vivo (Fig. 1c).

An explanation for this resistance phenotype could be that the TR cell populations had lost the expression of either the antigen-presenting HLA molecule or the MART-1 antigen itself, both of which would result in loss of TCR T cell recognition. However, HLA-HLA-A2 levels remained expressed in all TR cell lines, similar to parental cell lines (Supplementary Fig. 1b, c). In contrast, we observed reduced expression of the MART-1 protein in the four out of six TR cell lines (Fig. 1d, Supplementary Table 1 for primer sequences). This prompted the question whether this resistance phenotype was caused by antigen-low tumor cell selection or accompanied by, for example, a change in cell state. Therefore, we assessed whether also the T cell response towards other TCR antigens, which as opposed to MART-1 are unrelated to melanocyte differentiation, was altered in the TR melanoma cells. To bypass the potential confounder of differences in antigen expression, we exogenously loaded one of two different peptides (CDK4[R24C] and NY-ESO-1) on melanoma cells, and subsequently co-cultured them with CD8+ T cells that had been transduced with their cognate TCRs. Whereas the parental melanoma cell lines displayed dose-dependent sensitivity to these T cells upon co-culture, the TR cells were significantly more resistant (Fig. 1e, Supplementary Fig. 1d). This was not explained by a lack of T cell activation, since the levels of secreted IL-2 in these co-cultures (a measure of T cell activation) were similar between parental and TR lines (Supplementary Fig. 1e). These results together indicate not only that the spontaneous acquisition of T cell resistance is commonly associated with downregulation of a TCR-recognized melanoma antigen, in line with previous reports[19,20], but also that this is coupled to a more fundamental cellular change that renders melanoma cells resistant to cytotoxic T cells independent of the specific type of TCR : antigen interaction.

**NGFR[hi] melanoma cells pre-exist in tumors**. We previously reported that melanoma cells can acquire drug resistance through adopting specific phenotypic cell state changes, which are marked by elevated expression levels of the receptor tyrosine kinase AXL[14,15]. Also, melanomas can escape from differentiation

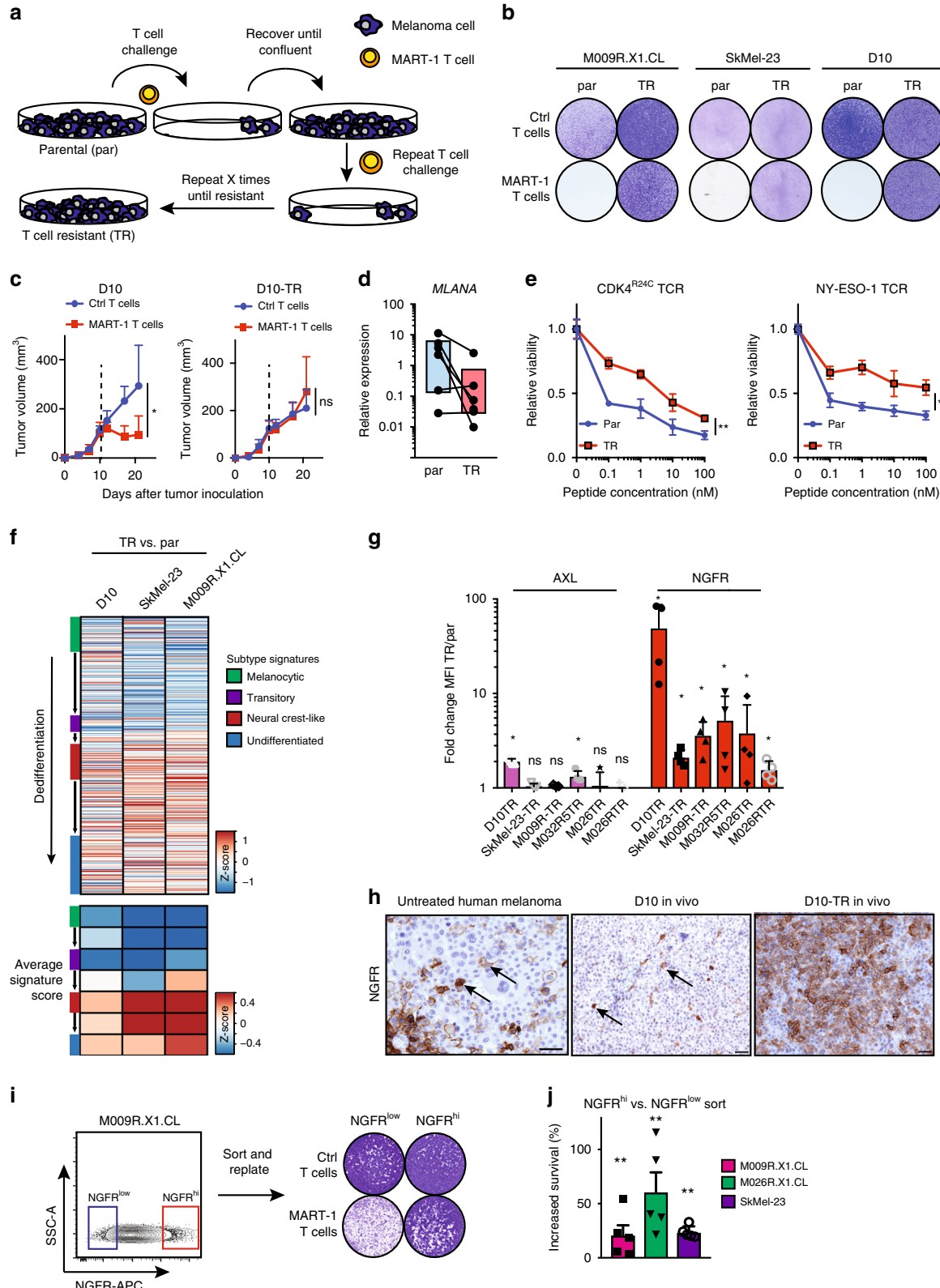

antigen-specific cytotoxic T cells through dedifferentiation, which is associated with an acquired and reversible induction of NGFR[19,20]. Therefore, we asked what the phenotypic cell state of the TR cells was compared with their parental counterparts. We performed RNA sequencing on three pairs of parental and TR cell lines and profiled them for several cell state markers, as described recently by Tsoi et al.[25]. All three cell lines showed molecular cell state changes upon the acquisition of a TR phenotype, and had shifted towards a predominantly neural crest-like cell state (Fig. 1f, Supplementary Table 2). At the protein level, NGFR upregulation appeared to be a shared hallmark compared with the parental counterparts, being upregulated 2 to 50-fold both for total MFI levels but also in terms of NGFR-positive fractions (Fig. 1g, Supplementary Fig. 1f, g). This was associated with

**Fig. 1 Melanoma fractions resistant to antigen-specific T cells. a** Graphic overview of the generation of T cell-resistant (TR) cell lines. **b** Colony formation assay of three parental cell lines (par) and their TR counterparts, treated with either control (Ctrl) or MART-1 T cells for 24 h in a 1:1 ratio and recovered for 3 days. Quantification in Supplementary Fig. 1a. **c** Tumor growth of melanoma cell lines D10 and the T cell-resistant counterpart, D10-TR. Randomization occurred on day 10 (indicated by dotted line). $n = 7$ per group except $n = 6$ for D10 + MART-1 T cells. Statistical analysis by Kruskal–Wallis test on day 21; $*p < 0.05$. **d** mRNA expression levels of *MLANA* in parental versus TR cell lines. Pooled data of six independent cell line pairs, lines indicate each paired parental and TR cell line. One experiment of three pooled technical replicates is shown; the data are reproduced in two independent replicates (available in Source data). **e** Cell viability after T cell attack of M009R.X1.CL cells for CDK$^{R24C}$ and NY-ESO-1 TCRs. An experiment of two independent replicates with three technical replicates is shown (other replicate can be found in Source data). Statistical analysis by unpaired *t*-test; $**p < 0.01$. **f** Gene expression changes and average signature scores in TR cells vs. parental cells, normalized per cell line. Gene lists for different subtypes/cell states were derived from Tsoi et al. (see Methods section). **g** Flow cytometry quantification of AXL and NGFR expression in six matched cell lines. Error bars indicate S.D. of four independent replicates. Statistical analysis by Mann–Whitney test; $*p < 0.05$, ns not significant. **h** Immunohistochemistry analysis of NGFR on human melanoma, D10 tumors and D10-TR tumors. Scale bars indicate 50 μm. **i** FACSort lay-out of NGFR$^{low}$ and NGFR$^{hi}$ cells. **j** Quantification of cell viability after MART-1 T cell attack in three cell lines, similarly sorted as in **i**. Data of two independent replicates with two and three technical replicates, respectively. Statistical analysis by Mann–Whitney test; $*p < 0.05$, $**p < 0.01$. All error bars in this figure represent S.D. Source data are provided as a Source Data file.

increased *SOX10* expression in most TR cell lines, another marker of the neural crest phenotype[23] (Supplementary Fig. 1h). In contrast, AXL was upregulated only mildly, and only in two out of six TR cell lines, excluding this as a frequent event (Fig. 1g).

The results above raise the possibility that NGFR$^{hi}$ cells constitute a therapeutically relevant melanoma subpopulation, which is associated with a selective advantage in the context of T effector cells. Such tumor fractions can be induced reversibly on immunotherapy, as has been shown previously[19,20]. From a clinical point of view, it would also be of interest to determine whether NGFR$^{hi}$ cells pre-exist as rare melanoma subpopulations, marking a pool of intrinsically treatment-resistant cells. We therefore assessed whether NGFR$^{hi}$ tumor cells can be detected already in untreated human melanomas. We analyzed by immunohistochemistry (IHC) a panel of clinical samples derived from untreated patients. We observed that nine out of 17 (52.9%) tumors contained melanoma cells expressing NGFR, with percentages ranging from 1 to 100% (median 10%) (examples in Fig. 1h, quantification in Supplementary Fig. 1i). This was recapitulated in a transplanted human melanoma cell line (D10) in mice: whereas parental D10 tumors harbored only rare NGFR$^{hi}$ cells, they accounted for the majority in D10-TR tumors (Fig. 1h). These analyses indicate that both melanomas in patients and human melanoma cell lines grown as xenograft tumors harbor NGFR$^{hi}$ cells prior to any treatment.

We observed that initially only small fractions of cells survived T cell attack and that those selectively expanded as a function of multiple challenges. Because of this finding and the observations above, we next asked whether patient-derived and standard established melanoma cell lines contain pre-existing NGFR$^{hi}$ melanoma cells, and if so, whether they are less susceptible to T cell elimination. FACS analysis identified both NGFR$^{lo}$ and NGFR$^{hi}$ cells, which were subsequently sorted to assess their relative T cell sensitivities. Tumor cells harboring high cell surface expression of NGFR were much more resistant to MART-1 T cells than the NGFR$^{lo}$ population, as judged by a co-culture killing assay (Fig. 1i, j). This was not caused by different levels in antigen expression (Supplementary Fig. 1j). Together, these results suggest that NGFR$^{hi}$, neural crest-like melanoma cells pre-exist in patients and that, at least in vitro and upon transplantation in mice they are in a distinct cellular state that is associated with resistance to T cell antitumor activity.

**NGFR$^{hi}$ melanomas are resistant to multiple therapies**. For AXL$^{hi}$ tumor cells, we previously reported that they are resistant not only to BRAF inhibition but also to inhibition of MEK or the combination[14,18]. To characterize NGFR$^{hi}$ melanoma cells in a similar way, we first investigated if they showed resistance to any key T cell cytokine. We focused on interferon-gamma (IFNγ) and TNF, since it is well established that specific tumor signaling

pathways determine the susceptibility to these cytotoxic and immunogenic T cell factors[29–32]. While these cytokines had a cytotoxic effect on the parental cell lines, their TR counterparts survived significantly better (Fig. 2a, b, Supplementary Fig. 2a). These results suggest that TR cells show considerably reduced susceptibility to both IFNγ and TNF. Since these cytokine signals are relayed by independent tumor pathways, these results support the notion that TR cells are in a different state, which alters their susceptibility to different cytotoxic agents.

To further investigate this phenomenon, we assessed any cross-resistance of TR melanoma cells towards other clinically relevant treatments. First, we tested whether BRAF and MEK inhibitors, which are commonly used treatment modalities for BRAF mutant melanoma patients, show differential killing of parental versus TR cells. The sensitivity to these compounds was significantly compromised in multiple TR cell lines compared with their parental counterparts, including BRAF + MEK inhibitor sensitivity in the BRAF$^{V600E}$ cell line D10 and the PDX cell line M009R.X1.CL, and MEK inhibitor sensitivity in the BRAF$^{WT}$/NRAS$^{WT}$ cell line SkMel-23 (Fig. 2c, d). This could not be ascribed to reduced proliferative activity in the TR cell lines (Supplementary Fig. 2b). To test any consequential selective advantage in a longer time span, we set up an assay of paired parental and TR cells that were exposed to a high concentration of BRAF inhibitor in vitro. Although this drug initially caused similar inhibition of tumor cell proliferation comparing parental and TR cells, only TR cells resisted prolonged BRAF inhibition (Fig. 2e). Dacarbazine is a chemotherapy modality that was used to treat melanoma patients prior to the introduction of precision medicines, while irradiation can be used to treat melanoma locally. TR cell lines showed also modestly reduced sensitivity to both these therapies compared with parental cell lines (Supplementary Fig. 2c, d).

Finally, we asked whether NGFR expression can serve as an independent predictor for lack of response to BRAF(+MEK) inhibition. Analysis of our cohort of 95 melanoma PDX, several of which were derived from BRAF inhibitor-relapsed tumors, showed that NGFR expression levels were significantly higher in the BRAF inhibitor-resistant melanomas (Fig. 2f). This was validated using NGFR IHC in a clinical dataset comprising samples from patients treated with BRAF + MEK inhibition, demonstrating that NGFR expression is a significant predictor of nonresponse (Fig. 2g). In conclusion, these results show that NGFR$^{hi}$ TR melanoma cells display general cross-resistance to both a variety of anticancer drugs and various types of immune pressure, suggesting that these cells are in a state of multi-drug- and T cell-resistance.

**Pre-existing NGFR$^{hi}$ cells display a stable phenotype**. We previously showed that not only AXL expression is acutely induced by BRAF + MEK inhibition, but also that AXL$^{hi}$ cells can pre-exist and selectively enriched upon BRAF inhibitor

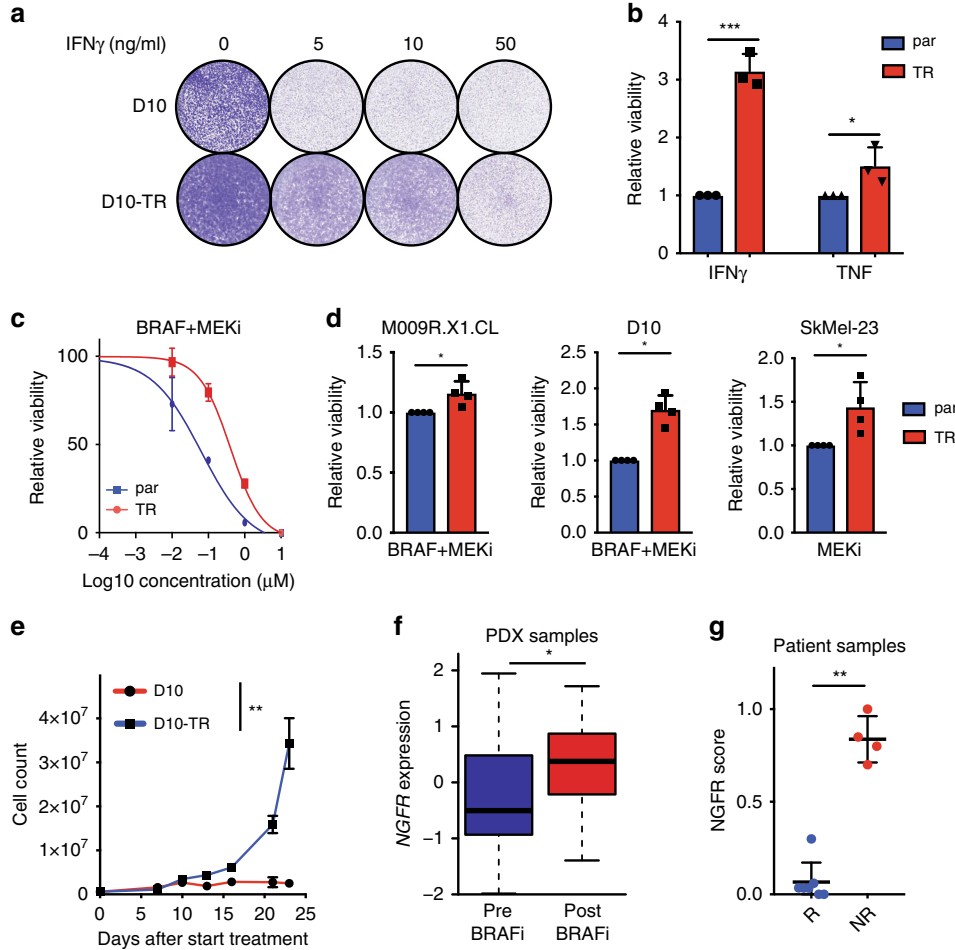

**Fig. 2 NGFR^hi melanomas are resistant to multiple therapies. a** Colony formation assay on IFNγ treatment on D10 and D10-TR cells. Tumor cells were treated for 7 days with IFNγ (which was refreshed on day 4) and stained with crystal violet. Quantification in **b**. **b** Quantification of IFNγ and TNF treatment for 7 days on D10 and D10-TR cells (medium was refreshed on day 4). Error bars represent S.D. of three independent replicates. Statistical analysis by unpaired $t$-test; *$p < 0.05$, ***$p < 0.001$. **c** Cytotoxicity assay in D10 parental versus TR cells for BRAF + MEKi. Titrations were performed with a 10:1 ratio of BRAFi:MEKi concentration. Error bars represent S.D. of three biological replicates; the experiment was performed in four independent replicates (as quantified in **d**). **d** Quantification of three cell lines for BRAF/MEKi sensitivity in parental versus TR lines. BRAF inhibitor dabrafenib was given at 10 nM and MEK inhibitor trametinib at 1 nM. Error bars represent S.D. of four independent replicates. Statistical analysis by Mann–Whitney test; *$p < 0.05$. **e** Cell count in parental and TR cells of the D10 cell line, treated continuously with 1000 nM of dabrafenib. Error bars represent S.D. of two independent replicates. Statistical analysis by Mann–Whitney test; *$p < 0.05$. **f** Average NGFR expression in PDX samples from melanoma. Error bars represent S.D. Statistical analysis by unpaired $t$-test. *$p < 0.05$. **g** NGFR immunohistochemistry score for patient samples prior to and on BRAF + MEK inhibition, divided in responders (R) and nonresponders (NR). Statistical analysis by Mann–Whitney, **$p < 0.01$. Source data are provided as a Source Data file.

pressure[18]. We examined, therefore, what the dynamics of NGFR expression are in the TR cells. It has previously been reported that NGFR can be reversibly induced by cytokines such as TNF, as well as by T cells[19,20]. Consistent with this, we observed that in parental (largely NGFR^lo) melanoma cells, NGFR was induced by T cells (independent of their TCR specificity), to return to baseline levels 3–14 days later (Fig. 3a, Supplementary Fig. 3a).

In contrast, pre-existing NGFR^hi cells, which were sorted from a polyclonal pool of (patient-derived) cell lines (Fig. 1h, i), displayed a stable phenotype of high NGFR expression, which they maintained for at least 4 weeks after sorting (Supplementary Fig. 3b). Similarly, also TR cells showed a stable NGFR^hi expression state that was maintained over multiple months (Fig. 3b). This was the case also at the single-cell level, since single-cell clones from a TR cell line were all stably resistant to T cells (Fig. 3c). Moreover, all TR clones were still NGFR^hi compared with D10 parental clones, indicating that even at a clonal level the NGFR-status is maintained (Supplementary Fig. 3c). Also the in vivo establishment of T cell-resistant tumors remained stable over multiple passages in mice

(Supplementary Fig. 3d, e). These results are consistent with, and extend previous results[18–20] demonstrating that, similarly to what we reported for AXL in the context of BRAF + MEK inhibition, NGFR can be both acutely induced upon treatment and expressed stably at high levels in pre-existing fractions.

Since we observed that pre-existing NGFR^hi tumor cells could expand selectively on T cell pressure, and because we observed that those expanded fractions are resistant to BRAF + MEKi, we then tested whether this cross-resistance phenomenon also occurred in vivo. First, we treated melanomas with MART-1-specific or control T cells. Acquired resistant D10 tumors expressed high levels of NGFR (Supplementary Fig. 3f). We sequentially treated these tumors with BRAF + MEK inhibitors and compared the response to tumors that were treated with control T cells. Whereas the latter group showed tumor reduction upon BRAF + MEK inhibitor treatment, tumors that acquired T cell resistance appeared cross-resistant to MAPK pathway inhibition (Fig. 3d). We conclude from these data that pre-existing NGFR-expressing cells can enrich on treatment in vivo to form a stable pool of cross-resistant tumor cells.

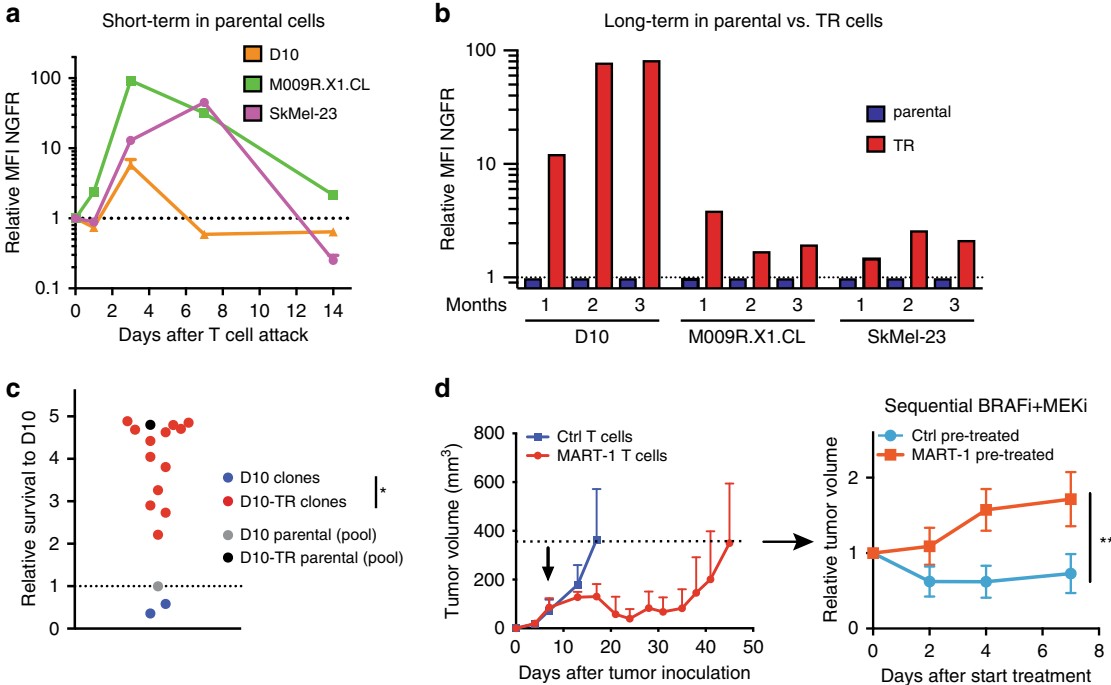

**Fig. 3 Pre-existing NGFR$^{hi}$ cells display a stable phenotype. a** Relative MFI as assessed by flow cytometry for parental cell lines after a 24 h co-culture with MART-1 T cells in a 1:8 ratio. After 24 h, T cells were washed off and the medium was refreshed every 3 days. Data shown from three independent replicates. **b** Relative MFI as assessed by flow cytometry of TR cell lines compared with their parental counterparts over a period of 3 months without any T cell stimulation. **c** D10 and D10-TR clones were obtained by FACS-based single cell sorting and expansion, and after 2 months they were subjected to MART-1 T cells in a 1:1 ratio. The relative survival compared with parental D10 cells is shown, based on quantification of the corresponding colony formation assays (from two independent replicates). Statistical analysis by Mann–Whitney test by comparing all TR clones to all parental clones; *$p < 0.05$. **d** D10 tumors were treated at day 7 with Ctrl or MART-1 T cells in vivo, and at an average tumor volume of 400 mm$^3$ individual mice were re-stratified to sequential BRAF + MEKi treatment ($n = 6$ per group; dabrafenib 30 mg/kg, trametinib 0.1 mg/kg). Error bars represent S.D. Statistical analysis by Mann–Whitney test on day 7; **$p < 0.01$. Source data are provided as a Source Data file.

**NGFR predicts immunotherapy resistance in melanoma patients**. Above, we showed that NGFR expression is a predictor of nonresponse to BRAF + MEK inhibition. Given our observation that high NGFR expression is also associated with T cell resistance, we determined whether this could be extended to an immunotherapy setting. We addressed this by investigating RNA profiling datasets of patients treated with immune checkpoint blockade[33,34]. We observed that NGFR expression at baseline was indeed higher in tumors that did not respond to anti-PD-1 and/or anti-PD-1 + anti-CTLA-4 combination therapy relative to those that did, and was further increasing on treatment in these patients (Supplementary Fig. 4a).

To determine whether an NGFR-associated cell state would have predictive value in a clinical setting, we first developed a tumor-intrinsic NGFR signature of melanoma, using RNA sequencing data derived from our melanoma PDX-platform[27]. We identified genes that were specifically upregulated in NGFR$^{hi}$ melanoma PDX compared with NGFR$^{lo}$ ones, and combined them to generate a tumor-intrinsic NGFR signature (Fig. 4a, Supplementary Fig. 4b). Then, we performed RNA sequencing on three matched parental and TR melanoma cell line pairs and determined whether the NGFR signature was enriched in the TR cells, which was indeed the case (Fig. 4b). This prompted us to investigate whether this signature correlated to immunotherapy response in melanoma patients. We applied the gene signature using Gene Set Enrichment Analysis (GSEA) to two gene expression cohorts of patients prior to start of anti-PD-1 treatment[33,34]. For both datasets the signature was significantly associated with higher expression in nonresponders (Fig. 4c, d). This was confirmed by IHC in a clinical dataset comprising

responders and nonresponders to anti-PD-1 therapy, in which we found a significant enrichment for NGFR$^{hi}$ tumors in non-responding patients (Supplementary Fig. 4c). These results indicate that tumor-intrinsic NGFR expression can predict T cell resistance in cell fractions, both in vitro and in patients.

**NGFR$^{hi}$ melanoma fractions show immune exclusion in patients**. Based on our finding above that an NGFR genetic signature predicts resistance to T cell killing, we also investigated the composition of melanomas in relation to their NGFR expression in situ. We therefore analyzed a dataset of untreated melanomas from patients using IHC of NGFR. We observed a significant anticorrelation between NGFR expression in melanoma cells and the presence of tumor-infiltrating T cells, characterized by CD3 expression and CD8 expression (Fig. 5a–c). Even within tumors that were heterogeneous for NGFR, we observed an anticorrelation between T cell infiltrates and melanoma NGFR expression (Fig. 5d). Computational estimation of T cell infiltrates in NGFR$^{lo}$ vs. NGFR$^{hi}$ melanomas in TCGA using MCP counter showed an opposite pattern (Fig. 5e). This apparent discrepancy between bulk RNA sequencing and immunohistochemistry analyses underscores the importance of including spatial information (Fig. 5a–d) for complete interpretation of these types of analyses, as also others have noted[19,35]. This association between T cells and NGFR$^{hi}$ melanoma cells may be caused by the initial requirement of T cells to establish an NGFR$^{hi}$, T cell-resistant (TR) tumor population before being excluded by as yet to be identified mechanisms. Collectively, these results suggest that NGFR$^{hi}$ melanoma cells are not only in a T cell-resistant cell state but are also associated with poor T cell infiltration.

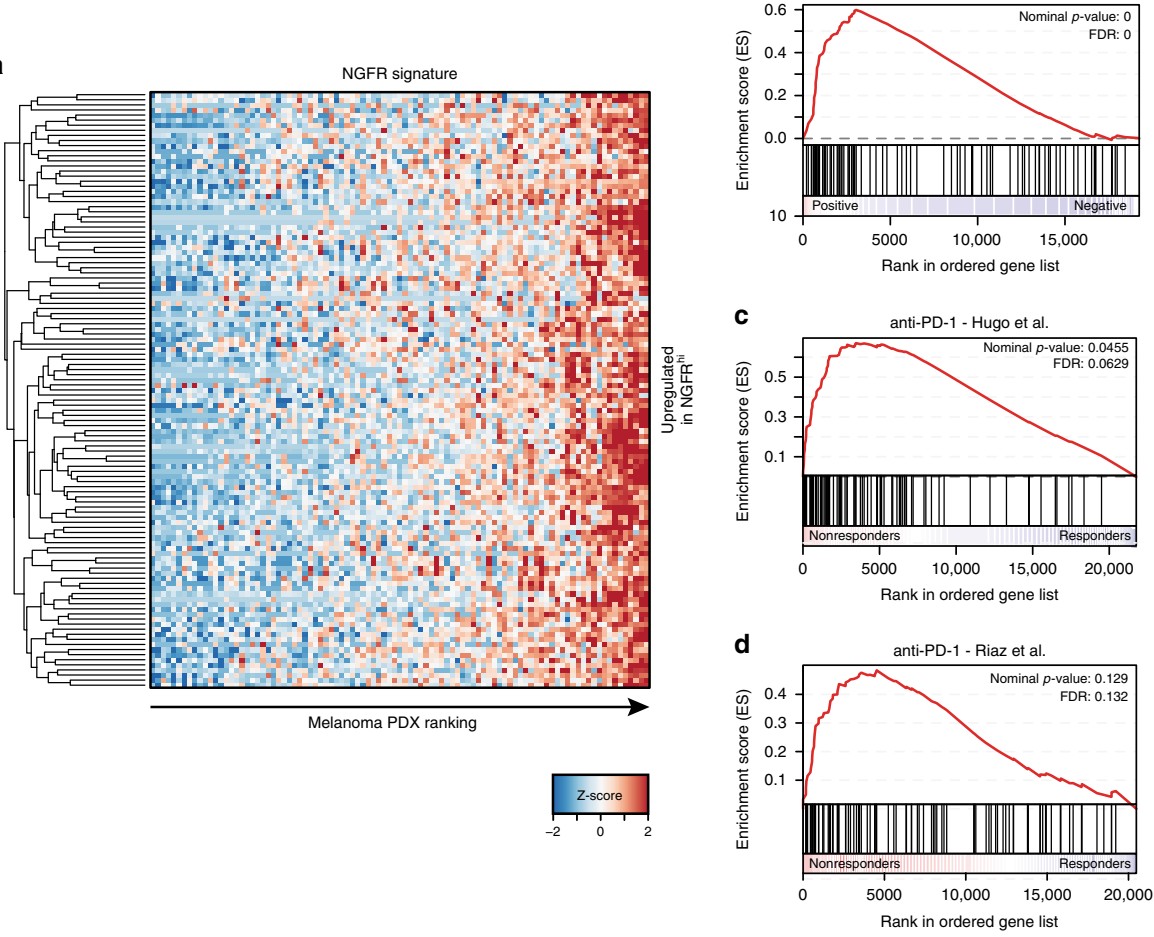

**Fig. 4 NGFR predicts immunotherapy resistance in melanoma patients. a** Heat map of the NGFR signature on a cohort of $n = 95$ melanoma PDX. **b** Gene Set Enrichment Analysis (GSEA) of NGFR signature ("upregulated in NGFR[hi]") on three parental vs. TR cell lines. **c, d** GSEA on two clinical datasets of patients treated with anti-PD-1 therapy. References for the source data of these databases can be found in the Methods section.

**Genetic perturbation of NGFR modulates T cell sensitivity**. All of the results above prompted the question as to whether NGFR functions only as a prognostic and predictive biomarker or whether it actually serves as a causal factor in T cell resistance. Therefore, we transduced multiple TR lines with shRNAs targeting NGFR. The two most effective hairpins led to a profound reduction in NGFR levels (Fig. 6a). These NGFR knockdown cell lines were significantly more susceptible to T cell antitumor activity than control cell lines, indicating that NGFR indeed plays a causal role in T cell resistance (Fig. 6b, Supplementary Fig. 5a, b). Of note, this effect was not observed in the (NGFR-low) parental cells (Supplementary Fig. 5c).

Given that NGFR depletion restores T cell-induced sensitivity, we investigated whether the reverse also holds true. We ectopically expressed NGFR in parental cells and assessed their T cell sensitivity compared with control cells. Overexpression of NGFR was sufficient to drive T cell resistance in both cell lines examined, indicating that NGFR is a critical factor contributing to T cell sensitivity (Fig. 6c, d).

Then, we determined how NGFR expression on the surface of TR cells mechanistically contributes to T cell resistance. First, we observed that knockdown of NGFR promoted caspase-dependent apoptosis upon T cell encounter, suggesting that NGFR protects against pro-apoptotic signaling upon T cell exposure (Supplementary Fig. 5d). These findings were recapitulated with NGFR knock-out cell lines generated by CRISPR-Cas9 (Supplementary

Fig. 5e–f). We subsequently asked whether the ligands that promote NGFR signaling, a family of neurotrophins, contribute to the observed phenotypes. TR cells produced significantly higher levels of BDNF, a neurotrophic factor structurally related to NGF (Fig. 6e). This prompted the question as to whether BDNF expression in TR cells contributed to the T cell resistance phenotype. BDNF knockdown phenocopied the effects of NGFR knockdown, and promoted T cell sensitivity in TR cells (Fig. 6f, Supplementary Fig. 5g). Together, these findings indicate that increased BNDF and NGFR expression together contribute to T cell resistance.

**Pharmacological reversal of NGFR[hi] state restores T cell sensitivity**. The finding that genetic perturbation of NGFR restored T cell sensitivity led us to determine whether pharmacological downregulation has similar effects. First, we focused on an inhibitor, which has been reported to block NGFR (AG-879[36]). Indeed, this compound was able to resensitize multiple TR lines to T cell killing (Fig. 7a, b), with little cytotoxicity (Supplementary Fig. 6a). Of note, this effect was largely diminished in NGFR-depleted cells, indicating that the drug acts on NGFR signaling (Supplementary Fig. 6b).

In view of this finding, we wished to expand our findings to a therapeutically more relevant compound. We focused on HSP90 inhibitors, a clinically tested class of compounds which has been

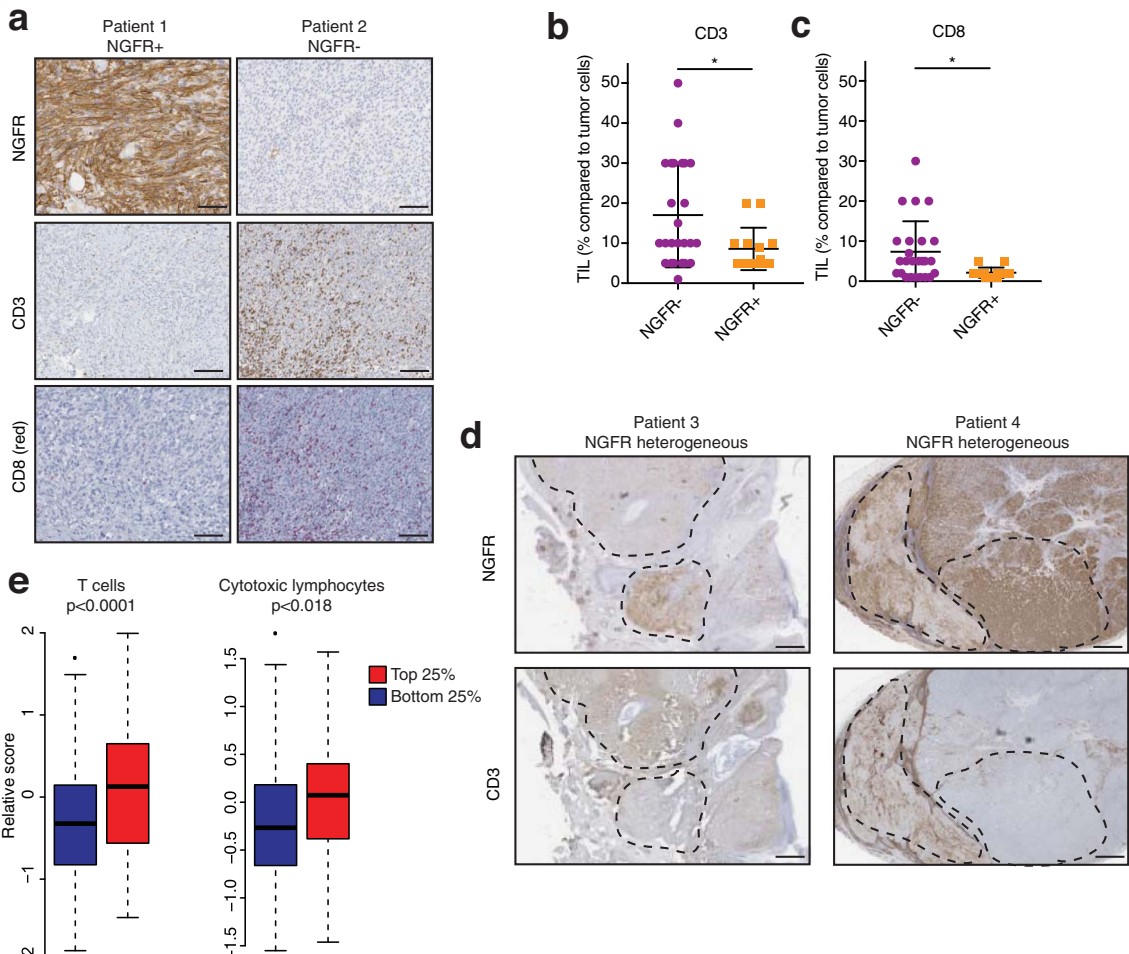

**Fig. 5 NGFR^hi melanoma fractions show immune exclusion in patients. a** Examples of two melanomas from patients showing either high or low NGFR expression, and the corresponding CD3 and CD8 stainings for the same regions. Scale bars indicate 50 μm. Quantification in **b**, **c**. **b**, **c** CD3 and CD8 scores for NGFR^neg vs. NGFR^pos samples. Only tumor-infiltrating lymphocytes were scored and compared with the relative abundance of tumor cells. Error bars represent S.D. Statistical analysis by Mann–Whitney test; *$p < 0.05$. **d** Examples of two melanomas from patients that are heterogeneous for NGFR expression within each sample, and the corresponding CD3 stainings for the same regions (indicated by dotted lines). Scale bars indicate 100 μm. **e** MCP counter analysis on melanoma cohort of TCGA divided in top and bottom 25% NGFR-expressing tumors. Gene sets for "T cells" and "cytotoxic lymphocytes" are plotted. Error bars represent S.D. Statistical analysis by unpaired *t*-test. Source data are provided as a Source Data file.

reported to block cytokine-induced EMT[8,9]. In melanoma (which are non-epithelial cells), dedifferentiation and phenotype switching resemble EMT[37], processes to which NGFR is known to contribute[24]. We first determined whether HSP90 inhibition changes NGFR expression in melanoma cell lines. For two different HSP90 inhibitors, ganetespib and 17-AAG, we indeed observed a dose and time-dependent reduction of NGFR in multiple TR lines (Fig. 7c, d, Supplementary Fig. 6c). To determine whether HSP90 inhibition can restore sensitivity to T cells, we co-cultured TR lines with or without T cells, in the presence of either of the two inhibitors. Whereas the TR lines were insensitive to T cells alone, treatment with HSP90 inhibitors substantially restored T cell sensitivity (Fig. 7e, f, Supplementary Fig. 6d, e). Notably, this was observed for T cells carrying one of multiple different TCRs, demonstrating that HSP90 inhibition restores immune sensitivity regardless of the tumor antigen (Fig. 7f). This effect was largely diminished in NGFR-depleted cells, indicating that ganetespib acts at least in part through NGFR downregulation (Supplementary Fig. 6f).

Finally, we tested whether these observations could be recapitulated in tumors in vivo. Mice bearing human TR melanoma xenografts were treated with two intraperitoneal injections of ganetespib (every 3 days) and tumors were harvested on day 6. At this time point, IHC confirmed that NGFR levels were downregulated in tumors (Fig. 7g), similarly to what we had observed in vitro. We proceeded to assess whether HSP90 inhibitors led to resensitization to T cell killing. We randomized tumor-bearing mice into groups inoculated with either control or MART-1 T cells, and with or without ganetespib. Whereas none of the single treatments caused a significant antitumor effect, ganetespib significantly restored the antitumor effect of MART-1 T cells (Fig. 7h). This allowed for a significant extension of overall survival of the mice receiving the combination treatment (Fig. 7i). These data indicate that HSP90 inhibitors downregulate NGFR and resensitize TR tumor cells to cytotoxic T cells in vitro and in mice, meriting clinical exploration of this treatment combination.

## Discussion

We report here that recurrent cycles of exposure to tumor differentiation antigen-specific cytotoxic T cells strongly select for a specific small fraction of immune-resistant melanoma cells, which pre-exists in patients and which is even stably maintained in established melanoma cell lines. This tumor

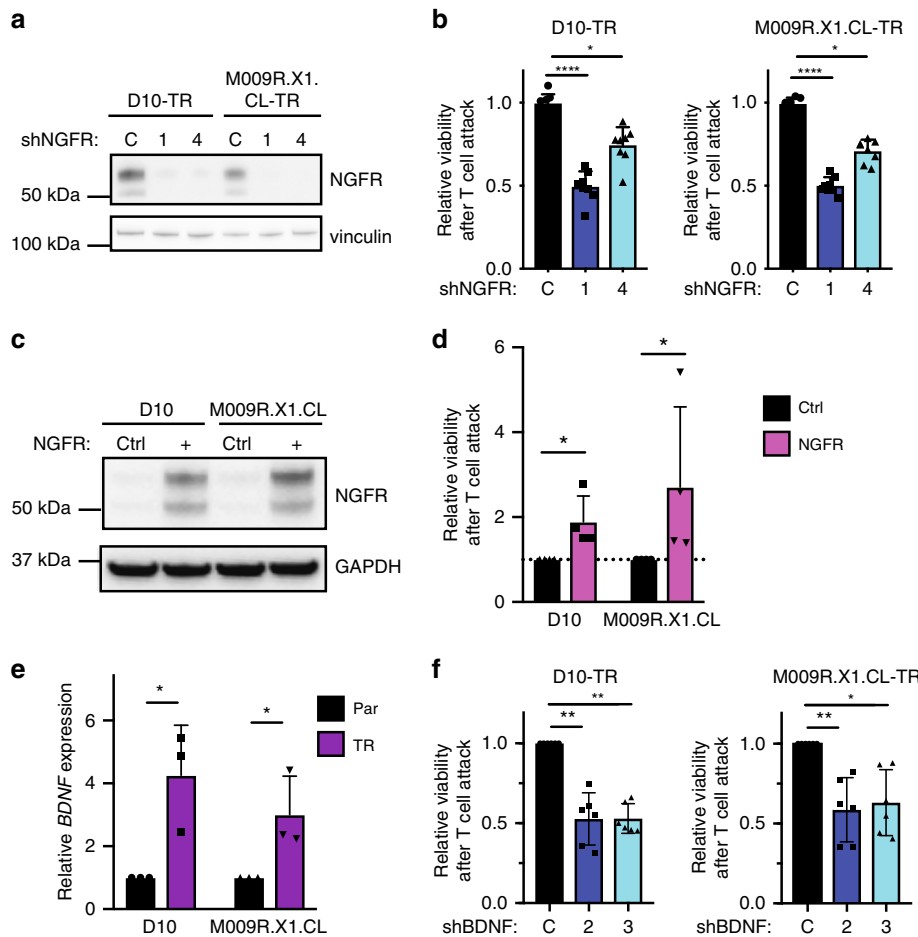

**Fig. 6 Genetic perturbation of NGFR modulates T cell sensitivity. a** Western blot analysis of indicated cell lines transduced with a control hairpin or one of two short hairpins targeting NGFR. Vinculin was used as a loading control. One blot of two independent replicates is shown (the other can be found in Source data). **b** Quantification of MART-1 T cell cytotoxicity in TR cell lines relative to Ctrl T cells (1:1 ratio tumor cell: T cell) in the presence of absence of shNGFR. Cells were loaded with 100 nM MART-1 peptide prior to the assay, and cytotoxicity was normalized to shCtrl cell survival after MART-1 T cell attack. Error bars represent S.D. of four independent experiments with two technical replicates. Statistical analysis by Kruskal–Wallis test; *$p < 0.05$, ****$p < 0.001$. **c** Western blot analysis on D10 and M009R.X1.CL cells transduced with a control ORF or an NGFR-RFP ORF. GAPDH was used as a loading control. One blot of two independent replicates is shown (the other can be found in Source data). **d** Quantification of cell survival by flow cytometry after T cell attack in vitro in control or NGFR-ORF cells. Error bars represent S.D. of four independent experiments. Statistical analysis by Mann–Whitney, *$p < 0.05$. **e** mRNA expression of *BDNF* in parental and TR cells. Error bars represent S.D. of pooled analysis of three independent replicates. Analysis by unpaired *t*-test; *$p < 0.05$. **f** Quantification of cytotoxicity in TR cell lines relative to shCtrl cells after T cell attack (1:1 ratio tumor : T cell) in the presence or absence of shBDNF. Error bars represent S.D. of six independent experiments. Statistical analysis by Kruskal–Wallis test; *$p < 0.05$, **$p < 0.01$. Source data are provided as a Source Data file.

subpopulation is characterized by high expression levels of the RTK NGFR but, unexpectedly, not AXL. NGFR[hi] cells are associated with intrinsic cross-resistance to T effector cells recognizing melanoma tumor antigens unrelated to differentiation, as well as with resistance to other clinically relevant therapies, including combinatorial BRAF + MEK inhibition. Further highlighting the clinical relevance of this group of tumor cells is the observation that in patients, they are associated both with T cell exclusion and resistance to immune checkpoint blockade (ICB). Lastly, we show that NGFR itself acts as a causal driver influencing melanoma susceptibility to cytotoxic T cells, and that, as a potentially beneficial consequence, this RTK could serve as a therapeutically tractable vulnerability.

Our results are consistent with, and extend, previous findings by several laboratories. First, it has been shown that in the vast majority of patient melanomas, NGFR-expressing cells lack expression of melanoma antigens, including MART-1[22,23]. Further, T cell cytokines such as TNF can induce NGFR

expression on tumor cells leading to the acquisition of MART-1/gp100 T cell resistance[19,20]. Consistent with those findings, we observed that in melanomas expressing no or low levels of NGFR, this RTK could be induced by T cells, to return to baseline levels shortly thereafter. However, we show that NGFR[hi] cells also commonly pre-exist in melanoma. Moreover, we demonstrate that NGFR[hi] melanoma fractions have a distinctive capacity to resist a variety of antigen-specific cytotoxic T cells, irrespective of whether their TCRs recognize a melanocyte differentiation antigen.

We previously showed that AXL[hi] states are associated with MAPK pathway inhibitor resistance. However, here we find that NGFR[hi] cell states, rather than cells characterized by high AXL expression, predict resistance to T effector cells. We also find that NGFR[hi] tumor cells are cross-resistant to BRAF (+MEK) inhibition, thereby corroborating other recent (single cell) approaches[13,25,38–40] and extending previous RNA profiles suggesting co-evolution of MAPKi resistance and CD8 T cell exhaustion[41].

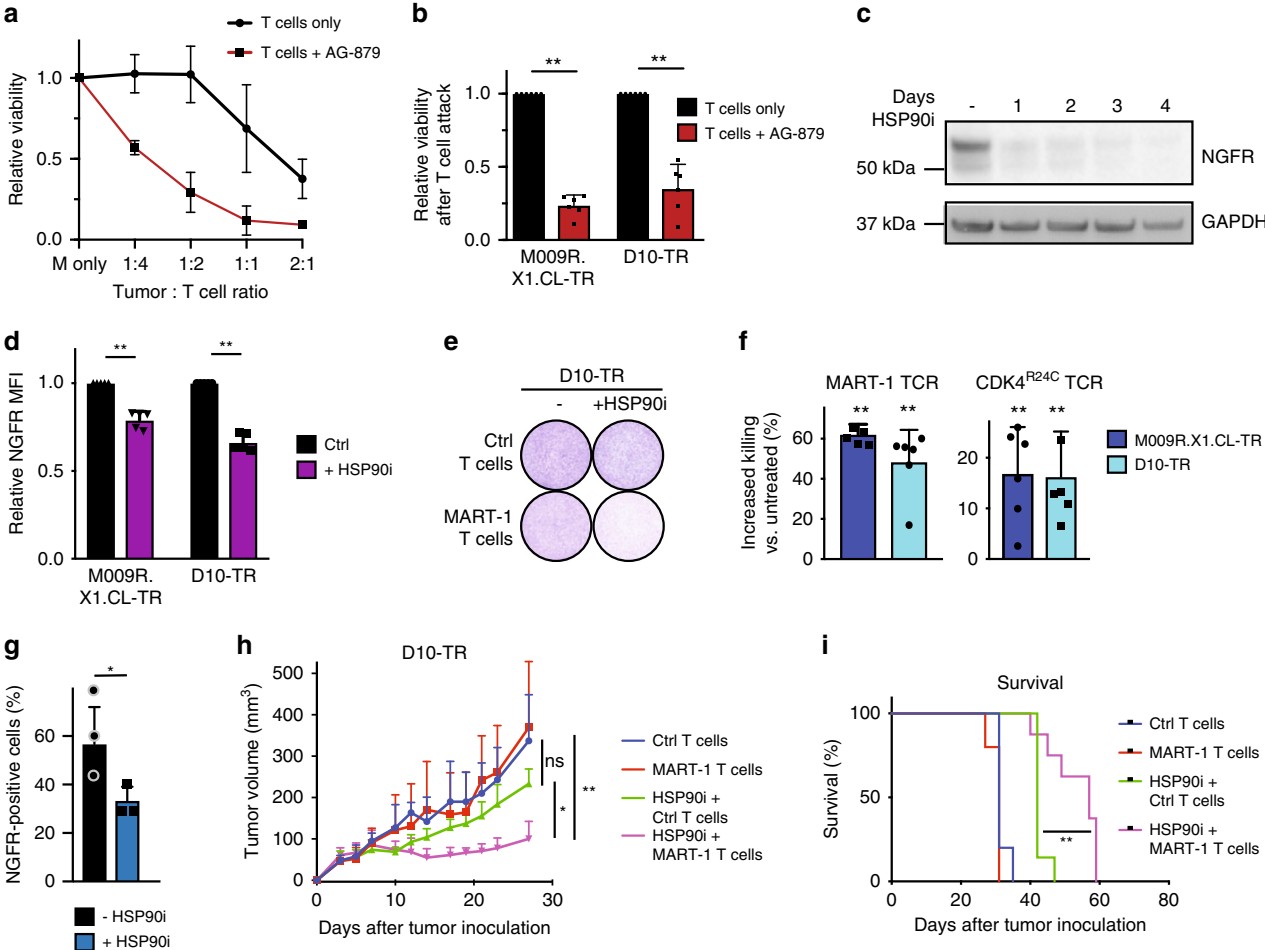

**Fig. 7 Pharmacological reversal of NGFR[hi] state restores T cell sensitivity. a** T cell titration in the presence or absence of tyrphostin (AG-879) in M009R.X1.CL-TR cells. **b** Quantification of MART-1 T cell cytotoxicity in TR cell lines relative to Ctrl T cells in the presence of absence of AG-879. Error bars represent S.D. of six independent experiments. Statistical analysis by Kruskal–Wallis test; **p < 0.01. **c** Western blot analysis on D10-TR cells treated with 250 nM of HSP90 inhibitor ganetespib for the indicated days. GAPDH was used as a loading control (see Source data for other replicate). **d** Quantification of NGFR expression by flow cytometry after 3 days of ganetespib treatment in vitro. Error bars represent S.D. of five independent experiments. Statistical analysis by Mann–Whitney, **p < 0.01. **e** Colony formation assay of D10-TR cells, treated with control (Ctrl) or MART-1 T cells in a 1:1 ratio and/or 250 nM ganetespib for 72 h. Quantification in **f**. **f** Quantification of cytotoxicity in TR cell lines relative to Ctrl T cells (1:1 ratio tumor cell: T cell) in the presence of absence of ganetespib. Two different sets of TCRs are shown. Error bars represent S.D. of six independent replicates. Statistical analysis by Mann–Whitney test; **p < 0.01. **g** Immunohistochemistry scoring of NGFR on D10-TR tumors after vehicle or 100 mg/kg ganetespib treatment in vivo during 6 days for n = 3 mice. Treatment was given on day 1 and day 4. Statistical analysis by unpaired t-test; *p < 0.05. **h** Tumor growth curve of D10-TR tumor cells after treatment with Ctrl T cells (n = 5), MART-1 T cells (n = 5), ganetespib 100 mg/kg + Ctrl T cells (n = 7), and ganetespib 100 mg/kg + MART-1 T cells (n = 8). Randomization, T cell injection, and start of ganetespib treatment occurred on day 7. Ganetespib was given twice weekly. Error bars represent S.D. Statistical analysis by Kruskal–Wallis test; ns not significant, *p < 0.05, **p < 0.01. **i** Survival curve belonging to **h**. Statistical analysis by Log-Rank Mantel–Cox test; **p < 0.01. Source data are provided as a Source Data file.

The dynamic behavior of NGFR expression levels as a function of T cell exposure seen by us here, by the Tüting lab in mice[19] and the Ribas lab in patients[20], is also in agreement with the recently reported role for this RTK in phenotype switching[24]. We demonstrate, however, that on top of this, there is a subpopulation of NGFR[hi] cells that pre-exists prior to treatment, in patients' melanomas, in PDX-derived melanoma cell lines and in commonly used established melanoma cell lines . These fractions show stable high expression of NGFR, which is maintained at least for several months. It therefore appears that NGFR is expressed in at least two different melanoma subpopulations: a group of NGFR-negative cells, in which it can be reversibly induced upon exposure to T cells or TNF, and as we demonstrate here, a subpopulation of stable NGFR[hi] cells, which exists in an (largely) irreversible state that is associated with both tumor and immune therapy resistance. In line with

this, ectopic expression of NGFR conferred T cell resistance to parental (NGFR-low) cells in vitro.

Corroborating this set of results with clinical data, we show that an NGFR gene expression signature predicts resistance to immunotherapy in melanoma patients, again in line with the idea that NGFR[hi] tumor cells represent a pool of ICB-refractory cells. Moreover, we show that NGFR[hi] tumor cell regions are largely devoid of T cells, which is seen both across melanomas and within heterogeneous melanoma areas. It is conceivable that this phenomenon contributes to the intrinsic immune resistance that these cells display, suggesting that NGFR[hi] cells are associated both with immune resistance and immune exclusion.

Lastly, given the remarkable resistance phenotype of NGFR[hi] melanoma cells, it was important to investigate any causal contribution of NGFR. Indeed, depletion of either NGFR or one of its neurotrophin ligands BDNF reverted T cell resistance.

Mechanistically, we observed that knockdown of NGFR promoted caspase-dependent apoptosis upon T cell encounter, suggesting that NGFR signaling protects against apoptotic signaling in this context. This could be pharmacologically recapitulated with HSP90 inhibition, which downregulates NGFR expression and contributes to T cell resensitization. HSP90 inhibition also reduces the expression levels of AXL[42], but because TR cells express NGFR but not AXL, we can rule out the latter as a critical mediator of this treatment. In conclusion, we propose that this study merits the development of NGFR-specific therapy, aiming to revert T cell- and ICB-resistance of melanoma.

## Methods

**Cell lines and cell culture conditions.** All melanoma cell lines, including PDX-derived[18] melanoma cell lines were obtained from the Peeper laboratory cell line stock. Fly cells were obtained from the Schumacher laboratory. Melanoma cell lines and HEK293T cells were cultured in DMEM (Gibco), and Fly cells were cultured in RPMI (Gibco), all with fetal bovine serum (Sigma), 100 U/ml penicillin and 0.1 mg/ml streptomycin (both Gibco) under standard conditions. Cell lines were regularly confirmed to be mycoplasma-free by PCR. HLA-A2 and MART-1 overexpression cell lines were generated through lentiviral transduction using one plasmid encoding HLA*02:01, MART-1, and Katushka[43]. HEK293T cells were used for virus production. They were transfected with the plasmid of interest, and the helper plasmids psPAX and pMD2.G (Addgene), using polyethylenimine as a transfection reagent. Viral supernatant was either snap frozen or immediately used for infection. Infected melanoma cells were sorted for Katushka-positive cells.

**Isolation and generation of TCR-specific CD8 T cells.** MART-1 (1D3) TCR retrovirus was produced in a packaging cell line[28], and supernatant was either directly used or snapfrozen for later use. NY-ESO and CDK4 TCR retrovirus was produced in Fly cells using polyethylenimine. Peripheral blood mononuclear cells were isolated from healthy donor buffycoats (Sanquin, Amsterdam, the Netherlands) by density gradient centrifugation using Lymphoprep (Stem Cell Technologies). CD8+ T cells were purified using CD8 Dynabeads (Thermo Fisher Scientific), activated for 48 h on a non-tissue culture treated 24-well plate that was pre-coated overnight with αCD3 and αCD28 antibodies (eBioscience, 16-0037-85 and 16-0289-85) at $2 \times 10^6$ per well. Activated CD8 T cells were harvested and mixed with TCR retrovirus and spinfected on a Retronectin coated (Takara, 25 μg per well) non-tissue culture treated 24-well plate for 2 h at 2000 g. After 24 h, T cells were harvested and maintained in RPMI (Gibco) containing 10% human serum (One Lamda), 100 units per ml of penicillin, 100 μg per ml of streptomycin, 100 units per ml IL-2 (Proleukin, Novartis), 10 ng per ml IL-7 (ImmunoTools), and 10 ng per ml IL-15 (ImmunoTools).

**Inhibitors, cytokines, and peptides for in vitro use.** MEK inhibitor GSK1120212/trametinib and GSK211436/dabrafenib were purchased from Selleck Chemicals (Houston, TX, USA). HSP90 inhibitors used were ganetespib (STA-9090, MedChemExpress) and 17-AAG (Bioconnect). The metabolic poison phenyl arsine oxide (PAO) and solvent dimethylsulfoxide were obtained from Sigma-Aldrich (St Louis, MO, USA). Dacarbazine was obtained from the Slotervaart hospital pharmacy. AG-879 was bought from MedChemExpress (HY-20878). All drugs except dacarbazine (which was already dissolved) were reconstituted in 100% dimethylsulfoxide to a final concentration of 1–10 mM. Recombinant human TNF (Peprotech) and IFNγ (R&D Systems) were diluted in sterile $H_2O$ to a final concentration of 100–200 μg/mL. Peptides for MART-1, CDK4, and NY-ESO-1 were reconstituted in 100% dimethylsulfoxide to a final concentration of 10 mM.

**Western blotting and antibodies.** Cell pellets were lysed in RIPA buffer (50 mM TRIS pH 8.0, 150 mM NaCl, 1% nonidet P40, 0.5% sodium deoxycholate, 0.1% SDS) supplemented with complete protease and phosphatase inhibitor cocktail (Thermo Fisher). Protein concentration was determined with the BCA Protein Assay Kit (Pierce). Western blotting was performed with standard techniques using 4–12% bis-tris polyacrylamide-SDS gels (NuPAGE, Life Technologies) and nitrocellulose membranes (Whatman, GE Healthcare). Blots were blocked in 4% milk in PBS plus 0.2% Tween 100 and incubated with primary antibody: NGFR (1:1000, #8238, CST) vinculin (1:10,000, V9131-100UL, Sigma) or GAPDH (1:1000, 1617002D09, Absea). The following secondary antibodies were used: goat anti-rabbit peroxidase conjugate (1:5,000, G21234) and goat anti-mouse (1:5,000, G21040), both purchased from Invitrogen. Western blots were incubated in a 1:1 dilution of solution 1 (0.1 M Tris pH 8, 2.5 mM luminol, 0.4 mM p-coumaric acid, all Sigma) and solution 2 (0.1 M Tris pH 8, 30% $H_2O_2$, all Sigma) and chemiluminescent signal was visualized using high performance autoradiography films (Hyperfilm MP, Amersham). Uncropped blots can be found in the Source Data.

**Flow cytometry.** For assessment of TCR transduction efficiency, the mouse TCR β chain (expressed within the construct) was stained (BD Pharmingen, 553172).

For melanoma analyses, cells were stained with HLA-A2-FITC conjugated antibody (1:50, 551285, BD), NGFR-APC (1:200, 345107, Biolegend), or AXL-PE conjugated antibody (1:200, FAB154P, R&D) for 30 min at 4 C and analyzed at LSRII or LSR Fortessa (BD Biosciences). For the NGFR and single-cell sort, melanoma cells were sorted on the FACSAriaII (BD Biosciences).

**Quantitative RT-PCR.** RNA was extracted using Isolate II RNA Mini Kit (Bioline) and 1 μg of total RNA was reverse transcribed into cDNA using Maxima First Strand cDNA kit (Thermo Scientific). Primers are listed in Supplementary Table 1. Real-time quantitative PCR amplification was performed using a 96-well plate system (Licor). Gene expression was normalized using the $\Delta\Delta$ Ct method, using RPL13 as a housekeeping gene.

**In vitro cytotoxicity assays.** For drug assays (without T cells), $2 \times 10^3$ cells were plated in 96-well plates and the drugs were added 1–3 h after seeding with the HP D300 Digital Dispenser (Tecan, Giessen, Germany). PAO was used as a positive control. After 3 days of incubation, the medium was replaced by a dilution of CellTiter Blue (Promega, Madison, WI) in medium. Fluorescence was measured by the Infinite M200 microplate reader (Tecan) after 3 h. The percentage of living cells was calculated using the following equation: % living cells = (signal treated − PAO)/(signal untreated − PAO) × 100%.

For T cell killing colony formation assays, $2 \times 10^5$ tumor cells were plated per well on a 6-well plate or $1 \times 10^5$ per well on a 12-well plate. CD8 T cells were admixed simultaneously in a 1:1 ratio (or other ratios if specified in figure and legends) and washed away after 24 h. After 3 days the plates were washed, fixed and stained for 1 h using a crystal violet solution containing 0.1% crystal violet (Sigma) and 50% methanol (Honeywell). For quantification, remaining crystal violet was solubilized in 10% acetic acid (Sigma). Absorbance of this solution was measured on an Infinite 200 Pro spectrophotometer (Tecan) at 595 nm. HSP90 experiments (ganetespib at 5 nM, 17-AAG at 25 ng/mL, co-treatment with T cells), and AG-879 experiments (10 μM pretreated tumor cells) experiments were performed in a 12-well assay: cells were plated $1 \times 10^5$ per well, peptide-loaded for 3 h and sequentially treated with T cells in different ratios for 3 days before read-out of the experiment. For peptide loading experiments (concentrations indicated in graphs), melanoma cells were pulsed for 3 h at the room temperature in a 96-well assay with the indicated peptides at 100 nM unless indicated otherwise and sequentially treated with T cells for 3 days before read-out of the experiment using CellTiter Blue (Promega, Madison, WI). For cell growth speed analysis (Supplementary Fig. 2b), Incucyte (Essen Bioscience) was used to calculate confluence over time.

**Animal studies and in vivo drug treatments.** Animal experiments were approved by the animal experimental committee (Instantie voor Dierenwelzijn) of the institute and performed according to Dutch law. All in vivo experiments were performed in male or female 8–12 week old NSG or NSG-β2M$^{null}$ mice (The Jackson Laboratory). Mice were inoculated subcutaneously at the right flank with $1 \times 10^6$ cells in 1:1 Matrigel (Corning) and normal DMEM medium. Tumor size was measured in a blinded fashion three times weekly with a caliper and tumor volume was calculated using the following formula: ½ × length (mm) × width (mm)$^2$. Randomization occurred in a blinded fashion, when tumors reached an average of 100 mm$^3$. Mice were treated intravenously with $5 \times 10^6$ untransduced or MART-1 specific T cells, diluted in 0.2 mL PBS per mouse. Dabrafenib (GSK211436) and trametinib (GSK1120212) were given daily orally at doses of 30 and 0.3 mg/kg, respectively, dissolved in 0.5% hydroxypropylmethylcellulose (Sigma), 0.2% Tween-80 (Sigma) in distilled water (HPMC) to a final volume of 300 μL/mouse. Ganetespib (MedChemExpress) was given intraperitoneally at a concentration of 100 mg/kg, every 3 days, after diluting in DMSO + Cremophor EL (2:1) with saline addition just before use (to a final concentration of 2:1:7). The experiment ended for individual mice either when the tumor size exceeded 1 cm$^3$, the tumor showed ulceration, in case of serious clinical illness, when the tumor growth blocked the movement of the mouse, or when tumor growth assessment had been completed. Differences in mean tumor volumes were compared between treatment groups using Kruskal–Wallis test. Mantel–Cox analysis of Kaplan–Meier curves was performed to analyze statistical differences in overall survival time with a general tumor size cut-off of 1000 mm$^3$.

**IHC on xenograft and patient samples.** The collection and use of human tissue was approved by the Medical Ethical Review Board of the Antoni van Leeuwenhoek. Informed consent was received from all patients for secondary use of tumor tissue. CD3 (RM-9107-S, Thermo Scientific) and CD8 (M7103, DAKO) stainings were performed using the Ventana autostainer using the standard protocol, and developed using brown and red visualization, respectively. NGFR was performed manually using the following protocols; slides were deparaffinized and antigen retrieval was performed using Tris/EDTA for 15 min in the pressure cooker. Slides were incubated with primary antibody (1:400, 8238, CST) overnight at 4 C. Secondary antibody was Polymer-HRP Anti-Rabbit Envision (K4011, Dako) and visualization was performed using DAB (Sigma). A counterstain with haematoxylin was performed and melanoma tissue slides were manually analyzed and scored by a certified pathologist. Only tumor cells were scored for NGFR expression, and only tumor-infiltrating lymphocytes were scored for CD3 and CD8 positive cells.

For the NGFR scoring in Fig. 2h, an intensity-weighted score was used by scoring for negative, +/1, 1+, or 2+ intensity. The following calculation was then used to establish the NGFR scoring: (%±) × 0.3+ (%1+) × 0.7+ (%2+) × 1.

**Knockdown/knockout of NGFR and BDNF.** HEK293T cells were transfected with two shRNA plasmids against the target of interest or a control hairpin, and the helper plasmids psPAX and MS2G (Addgene). For NGFR knockout experiments, sgRNAs targeting NGFR were cloned into lentiCRISPR-v2 (Addgene) and lentivirus was obtained using the same protocol as for shRNAs. Viral supernatant was either snap frozen or immediately used for infection. Infected melanoma cells were puromycin-selected. Sequences for shNGFR were: #1: 5′ GCACTGTAGTAA ATGGCAATT; #4: 5′ GACAACCTCATCCCTGTCTAT. Full hairpin sequences for shBDNF were: #2 5′ CCGGGAATTGGCTGGCGATTCATAACTCGAGTTA TGAATCGCCAGCCAATTCTTTTTG; #3: 5′ CCGGACAGTGGTTCTACAAT CTATTCTCGAGAATAGATTGTAGAACCACTGTTTTTTG. Guide sequence for sgNGFR was: TTGCAAGCAGGGCGCGAACG.

**Overexpression of NGFR.** HEK293T cells were used for virus production for NGFR overexpression vectors. HEK293T cells were transfected with plasmids encoding NGFR-RFP or a control vector, as described in Restivo et al.[24], and the helper plasmids psPAX and pMD2.G (Addgene). Viral supernatant was either snap frozen or immediately used for infection.

**RNA sequencing of TR cells.** RNA was isolated from the parental and TR cell lines as follows: cell pellets of $5 \times 10^6$ cells were homogenized in TRIzol reagent (15596-018, Ambion life technologies) using a polytron (DI 18 Disperser, IKA) in a 15 mL tube (Falcon) according to the manufactures protocol. Typically 1 mL of TRIzol reagent was used per 50–100 mg of tissue. Total RNA was extracted using TRIzol reagent (15596-018, Ambion life technologies) according to the manufactures protocol. Briefly, 0.2× volumes of chloroform (Chloroform stab./ Amylene, Biosolve) was added to the Trizol homogenate and the tube(s) (Falcon, 15 mL) were shaken vigorously. The tube(s) were incubated for 2–3 min at the room temperature and centrifuged (4500 RCF) (Hettich, rotanta 46 RS) for 1 h at 4 °C. Approximately 70% of the upper aqueous phase was transferred to a clean 15 mL tube and 0.5× volume of isopropanol (33539, Sigma-Aldrich) was added. The tube(s) were incubated overnight at −20 °C and centrifuged (4500 RCF) for 30 min at 4 °C. The supernatant was removed and the pellet was washed twice with 80% ethanol (32221-2.5 L, Sigma-Aldrich). The total RNA pellet was air-dried for 8 min and dissolved in an appropriate volume of nuclease free water (AM9937, Ambion life technologies) and quantified using Nanodrop UV–VIS Spectrophotometer. The total RNA was further purified using the RNeasy Mini kit (74106, Qiagen) according to the manufactures instructions. Quality and quantity of the isolated total RNA was assessed by the 2100 Bioanalyzer using a Nano Chip (Agilent, Santa Clara, CA) (also see Supplementary Data 1). >5000 ng of each sample was used for RNA sequencing and the RIN values were 9.8–10 for each sample. Strand-specific libraries were generated using the TruSeq Stranded mRNA sample preparation kit (Illumina Inc., San Diego, RS-122-2101/ 2, Illumina, Part # 15031047 Rev. E). Polyadenylated RNA from total RNA was purified using oligo-dT beads. Next, RNA was fragmented, random primed, and reverse transcribed using SuperScript II Reverse Transcriptase (Invitrogen, part # 18064-014) with Actinomycin D. Second strand synthesis was performed using Polymerase I and RNaseH with replacement of dTTP for dUTP, and fragments were 3′ end adenylated and ligated to Illumina Paired-end sequencing adapters and subsequently amplified by 12 cycles of PCR. Libraries were analyzed on a 2100 Bioanalyzer using a 7500 chip (Agilent, Santa Clara, CA), diluted and sequenced with 65 base single reads on a HiSeq2500 using V4 chemistry (Illumina Inc., San Diego). Sequence samples were mapped to the human genome (Homo.sapiens.GRCh38.v82) using STAR(2.6.0c) in two-pass mode with default settings. Because of the high quality RNA, removal of PCR duplicates was not needed and read counts were directly counted using HTSeq-count with default settings[44]. Normalization and statistical analysis of the expression of genes was performed using DESeq2[45]. Data can be found on GEO with accession code GSE147091.

**NGFR signature establishment in PDX samples.** RNA read count data of the melanoma PDX samples were downloaded from GEO (GSE129127). These samples were filtered for sequence reads due to possible mouse contamination using XenofilteR[46] and sequence reads were counted using HTSeq-count[44]. Normalization and statistical analysis of the expression of genes was performed using DESeq2[45]. Differential gene expression analysis was performed comparing the samples with the highest ($n = 16$) and lowest ($n = 16$) NGFR expression in the PDX dataset. Protein coding genes with an FDR < 0.01 and positive correlation with NGFR were maintained to comprise the NGFR gene signature ($n = 100$ genes).

**Patient RNA sequencing datasets.** Raw RNA sequence data (fastq files) of Fig. 4 were downloaded from SRA for the Hugo (PRJNA312948) and Riaz (PRJNA356761) datasets. The sequence reads were mapped to the human genome (Homo.sapiens.GRCh38.v82) using STAR(2.6.0c) with default settings. Sequence

reads were counted using HTSeq-count[44]. Normalization and statistical analysis of the expression of genes was performed using DESeq2[45]. RNA sequencing reads of Supplementary Fig. 4a) was provided by D.T.F., K.T.F. and G.B. RNA expression data for melanoma (SKCM, $n = 472$) samples were downloaded from the TCGA portal using the R-package 'TCGAbiolinks'. The downloaded count data was normalized using DESeq2[45] and the Z-score calculated (number of standard deviations below or above the population mean). The tool 'MCPcounter' (https:// github.com/ebecht/MCPcounter) was used to estimate the abundance of immune and stromal cells in the downloaded TCGA samples. MCPcounter was run on the 472 TCGA melanoma samples for all gene signatures as provided by MCPcounter (http://raw.githubusercontent.com/ebecht/MCPcounter/master/Signatures/genes. txt). For the gene signatures "T cells" and "Cytotoxic lymphocytes", boxplots were made based on the top 25% highest ($n = 118$) and 25% lowest ($n = 118$) NGFR-expressing samples. P-values were based on an unpaired t-test. The code used is available on GitHub https://github.com/PeeperLab/NGFR_NatComm2020.

**Gene Set Enrichment Analysis (GSEA).** GSEA was performed using the BROAD javaGSEA standalone version (http://www.broadinstitute.org/gsea/downloads.jsp) and the NGFR gene signature. Analysis were run using 10,000 permutations. Genes in the Hugo and Riaz datasets were ranked based on the Signal2Noise metric.

**Statistical testing.** The data of in vivo experiments were analyzed at the indicated time points in legends by non-parametric Mann–Whitney test for two conditions, or Kruskal–Wallis test when >2 conditions were compared. Survival analyses on Kaplan–Meier curves was analyzed using Log-Rank Mantel–Cox test. All analyses were performed using the Prism Graphpad software. The data of in vitro experiments on melanoma cell lines were analyzed using the Mann–Whitney test or unpaired t-test for two conditions or Kruskal–Wallis or one-way ANOVA test for >2 conditions, depending on whether or not a normal distribution was reached in the samples. Sample size was determined for mouse experiments on a power of 0.8, α 0.05 and estimated effect sizes + standard deviations using the program G*Power.

**Reporting summary.** Further information on research design is available in the Nature Research Reporting Summary linked to this article.

## Data availability
RNA read count data of the melanoma PDX samples were downloaded from GEO (GSE129127). Raw RNA sequence data (fastq files) were downloaded from SRA for the Hugo (PRJNA312948) and Riaz (PRJNA356761) datasets. The TR sequencing dataset is available on GEO (GSE147091). The analysis of Fig. 5e was performed on the TCGA melanoma database. The gene lists for the cell states/subtypes were taken from Tsoi et al.[25]. All relevant data are also available from the authors. The source data underlying the figures are provided as Source Data files. Source data are provided with this paper.

## Code availability
Codes that were used are XenofilteR[46], HTSeq-count[44], and DESeq2[45]. Source data are provided with this paper.

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

## Acknowledgements

We thank all the members of the Peeper and Blank laboratory for their valuable input and the FACS facility and animal facility at the NKI for their support. We would like to acknowledge the NKI-AVL Core Facility Molecular Pathology & Biobanking (CFMPB) for supplying NKI-AVL Biobank material and lab support. We thank Mireille Toebes, Wouter Scheper and Ton Schumacher for kindly providing the CDK4$^{R24C}$ and NY-ESO-1 TCR constructs, peptides and Fly cells, and Chong Sun for providing SOX10 primers. We thank Johanna Diener and Lukas Sommer for sharing the NGFR overexpression construct. The research leading to these results has been funded by the European Research Council under the European Union's Seventh Framework Programme (FP7/2007-2013)/ERC synergy grant agreement no. 319661 COMBATCANCER and grants NKI 2014-7241, NKI 2013-5799, and NKI 2017-10425 from the Dutch Cancer Society (KWF) to D.S.P.

## Author contributions

J.B., D.W.V., M.L., and D.S.P. designed experiments. J.B., D.W.V., M.L., B.B., J.K., M.P., and M.B. performed experiments. O.K. performed bioinformatic analyses and provided critical input. S.B., M.M., and A.v.A. collected patient samples that were stained for NGFR and immune markers, which were analyzed by J.B., S.B., J.-Y.S., and H.M.H. E.A.R., C.U.B., D.T.F., G.B., and K.T.F. provided clinical data and critical input. J.B. and D.S.P. wrote the paper. D.S.P. supervised the study.

## Competing interests

The authors declare the following competing interests: C.U.B. receives grants and/or research support from Novartis, BMS, and NanoString, and has received honoraria or consultation fees for MSD, BMS, Roche, Novartis, GSK, Pfizer, Lilly, Genmab, and Pierre Fabre. C.U.B. and D.S.P. are co-founders, shareholders and advisors of Immagene B.V. M.A.L. is co-founder, shareholder and CEO of Immagene B.V., unrelated to this study. K.T.F. has served on the Board of Directors of Clovis Oncology, Strata Oncology, Loxo Oncology, and Checkmate Pharmaceuticals; Scientific Advisory Boards of X4 Pharmaceuticals, PIC Therapeutics, Sanofi, Amgen, Asana, Adaptimmune, Fount, Aeglea, Shattuck Labs, Tolero, Apricity, Oncoceutics, Fog Pharma, Neon, Tvardi, xCures, Monopteros, and Vibliome; consultant to Lilly, Novartis, Genentech, BMS, Merck, Takeda, Verastem, Boston Biomedical, Pierre Fabre, and Debiopharm; and research funding from Novartis and Sanofi. G.M.B. has sponsored research agreements with Olink Pharmaceuticals, Palleon Pharmaceuticals, and Takeda Oncology. She has been on scientific advisory boards for Nectar Therapeutics and Novartis and served as a speaker for Novartis. All the other authors declare no competing interests.
