## [Peer Review File · Nature Communications]

Reviewers' comments:

Reviewer #1 (Remarks to the Author); expert on mouse models for melanoma and signaling:

This paper by Boshuizen and colleagues demonstrates a convincing correlation of high CD271/NGFR expression in melanoma cells with (MART-1) T cell resistance and resistance to other melanoma drugs. High CD271 expression has been associated with resistance to BRAF/MEK inhibition before. Likewise, a correlation of high CD271 expression with reduced T cell infiltration has previously been shown. Of note, the point crucial for the present study – inhibition of the NGFR^{high} state by blocking specifically CD271 (or e.g. by engineering CD271-specific T-cells) – has not been addressed and it remains open why CD271/NGFR^{high} cells are resistant to T cells.

Specific points:

1) For the key functional experiments, the authors use HSP90i, which is not at all a specific NGFR inhibitor. Downregulation of NGFR upon HSP90i might be merely correlative. Specific NGFR inhibition, at (or during acquisition of) the TR state in vitro and in vivo, has to be performed to assess whether inhibition of the CD271^{high} state indeed prevents relapse, i.e. whether resistance is indeed NGFR driven, as claimed in the title.

2) In Figure 1, the authors present a nice method for how to generate TR melanoma cells. However, the TR state is ill defined and it is unclear why the authors picked CD271/NGFR for their further studies (as it stands, the authors selected NGFR mainly based on literature; for this, the work shown in Fig1a-e would not have been necessary). What is the molecular signature of the TR state obtained in the experiments of Figure 1 and how does it compare to the signatures shown in Figure 4?

3) As shown by others before, resistance formation involves various cellular states rather than just one particular, homogeneous cell state. TR state analysis should optimally be performed at the single cell level. Rather than showing overall NGFR expression (Fig. 1 f), the numbers of CD271/NGFR^{high} cells as compared to CD271/NGFR^{low} cells have to be determined upon T cell resistance. Likewise, how many cells within the clones shown in Figure 3c are CD271 positive?

4) The choice of the cell lines used for a particular set of experiments is not consistent (e.g. Figure 1c vs 1h; different cell lines used for the different panels of Figure 2). Each key experiment should be repeated with at least two cell lines.

5) Why are CD271/NGFR^{high} cells resistant to T cells? The mechanisms underlying resistance formation have to be addressed.

Do CD271/NGFR^{low} cells play a role in this as well (e.g. by interaction with CD271/NGFR^{high} cells)?

6) In Figure 6b, does shNGFR already exhibit an effect in non-TR cells? This control experiment needs to be included.

7) Figure 5: what about NGFR expression in biopsies from resistant patients?

8) Minor point: in general, the labelling of figure axes could be improved; as an example, in Figure 3b, it's not clear immediately that NGFR MFI is measured, etc.

Reviewer #2 (Remarks to the Author); expert on PDX models, melanoma and immunotherapy:

This is a nice manuscript with some interesting findings. I have only three major concerns:

1. The immunotherapy experiments are not conducted with autologous tumor and TILs. Although HLA matched it is therefore risky that some of the effects are not general but only pertaining to allogeneic responses or regulation of antigen processing. The authors should consider repeating some of the data using paired autologous TILs and tumor cells.
2. The extended data 5 suggest that there might be an off target problem with the shRNA targeting NGFR. Please repeat the experiment with CRISPR/Cas9 deletion of NGFR.
3. Although the data are suggestive of the NGFR is necessary, which can be strengthened by CRISPR, is NGFR sufficient to drive resistance? If NGFR is transgenically overexpressed, will this result in resistance?

Reviewer #1 (Remarks to the Author); expert on mouse models for melanoma and signaling:

This paper by Boshuizen and colleagues demonstrates a convincing correlation of high CD271/NGFR expression in melanoma cells with (MART-1) T cell resistance and resistance to other melanoma drugs. High CD271 expression has been associated with resistance to BRAF/MEK inhibition before. Likewise, a correlation of high CD271 expression with reduced T cell infiltration has previously been shown. Of note, the point crucial for the present study – inhibition of the NGFR^{high} state by blocking specifically CD271 (or e.g. by engineering CD271-specific T-cells) – has not been addressed and it remains open why CD271/NGFR^{high} cells are resistant to T cells.

Author reply: We thank the reviewer for his/her enthusiasm regarding the data showing that NGFR-high tumor cells display resistance to several therapies, as well as for the thorough evaluation of the manuscript. In response to his/her comments, we have added several experiments to the manuscript that provide crucial insight into the mechanism by which NGFR drives immune resistance; please see detailed replies below. In total, we have included 7 new Main Figure Panels and 11 new Extended Data Figure panels.

Specific points:

1) For the key functional experiments, the authors use HSP90i, which is not at all a specific NGFR inhibitor. Downregulation of NGFR upon HSP90i might be merely correlative. Specific NGFR inhibition, at (or during acquisition of) the TR state in vitro and in vivo, has to be performed to assess whether inhibition of the CD271^{high} state indeed prevents relapse, i.e. whether resistance is indeed NGFR driven, as claimed in the title.

Author reply: We agree with the reviewer that HSP90 inhibitors are not NGFR-specific. We chose HSP90 inhibitors because they had been suggested to revert EMT-programs. This led us to hypothesize that, by analogy, they may also alter the TR state. Furthermore, these compounds are also of clinical relevance. However, we agree that more specific NGFR inhibition will strengthen our claim that NGFR is a driving factor in T cell resistance. Therefore, we have performed several experiments to answer the reviewer's question and strengthen this claim:

- a) We performed experiments using the NGFR inhibitor Tyrphostin AG-879^a. Tyrphostin treatment sensitized multiple NGFR-high TR cell lines to T cell-mediated cell killing, in line with the role of NGFR as a resistance factor (new **Figure 7a, b**). This effect was no longer observed in NGFR-depleted cells, demonstrating that the sensitization effect is NGFR-mediated (new **Extended Data Fig. 6a, b**).
- b) We included several new experiments to perturb, whether positively or negatively, the expression levels of NGFR. In addition to short hairpin-mediated downregulation we also performed CRISPR-mediated knock-out of NGFR (new **Extended Data Figure 5 e, f**), as well as ectopic expression of NGFR (new **Figure 6c, d**). The results from both knockout and ectopic expression studies corroborate our original data and demonstrate that NGFR is a key factor driving T cell resistance.
- c) We included a genetic epistasis experiment to determine the specificity of pharmacologic HSP90 inhibition. In both a regular and a PDX-derived melanoma cell line, the effects of HSP90 inhibitor ganetespib is largely diminished upon NGFR depletion (new **Extended Data Figure 6f**). These results further support the results above and show that ganetespib acts in least in part through NGFR.

^a <https://focusbiomolecules.com/ag-879-ngfr-kinase-inhibitor/>

Together, these new data strengthen our conclusion that NGFR serves as a key driver of T cell resistance, and that its pharmacological or genetic inhibition enhances T cell sensitivity.

2) In Figure 1, the authors present a nice method for how to generate TR melanoma cells. However, the TR state is ill defined and it is unclear why the authors picked CD271/NGFR for their further studies (as it stands, the authors selected NGFR mainly based on literature; for this, the work shown in Fig1a-e would not have been necessary). What is the molecular signature of the TR state obtained in the experiments of Figure 1 and how does it compare to the signatures shown in Figure 4?

Author reply: In response to the referee's remark we now included an RNA sequencing experiment comparing molecular signatures of TR vs. parental cells in three pairs of melanoma cell lines. In line with our original findings, TR cell lines have shifted towards a dedifferentiated cell state. They display a neural crest-like phenotype based on the recent classification as described by Tsoi *et al.* (Cancer Cell, 2018). We included these data in new **Figure 1f**. We noted some overlap between the Tsoi signatures and the NGFR-signature, and the overlapping genes (n=11) were all neural-crest/undifferentiated genes (see figure for Reviewer below).

Reviewer figure 1: overlap Tsoi and NGFR signatures.

3) As shown by others before, resistance formation involves various cellular states rather than just one particular, homogeneous cell state. TR state analysis should optimally be performed at the single cell level. Rather than showing overall NGFR expression (Fig. 1 f), the numbers of CD271/NGFR^{high} cells as compared to CD271/NGFR^{low} cells have to be determined upon T cell resistance. Likewise, how many cells within the clones shown in Figure 3c are CD271 positive?

Author reply: We thank the reviewer for this helpful suggestion. We have now plotted all fractions of the TR cells (new **Extended Data Figure 1g**) and the clones of Figure 3c (new **Extended Data Figure 3c**). This demonstrates that TR cells also have higher fractions of NGFR⁺ cells compared to parental cells, in addition to the overall NGFR expression.

4) *The choice of the cell lines used for a particular set of experiments is not consistent (e.g. Figure 1c vs 1h; different cell lines used for the different panels of Figure 2). Each key experiment should be repeated with at least two cell lines.*

Author reply: We made sure that every experiment was performed in at least two cell lines. Although examples of different experiments are sometimes shown from different cell lines, we ensured that the quantification of multiple cell lines is always plotted as well (usually in the **Extended Data Figures**), also for the newly added data.

5) *Why are CD271/NGFR^{high} cells resistant to T cells? The mechanisms underlying resistance formation have to be addressed. Do CD271/NGFR^{low} cells play a role in this as well (e.g. by interaction with CD271/NGFR^{high} cells)?*

Author reply: We agree that this is a key issue and therefore, in response to this comment we have performed several mechanistic experiments that provide critical new insight into the mechanism of NGFR-associated T cell resistance. Firstly, we expanded our findings on the role of NGFR as a resistance driver. We found that ectopic expression of NGFR is sufficient to cause T cell resistance in parental melanoma cells (new **Figure 6c, d**). Conversely, we show that, knockdown of NGFR promotes caspase-dependent apoptosis upon T cell encounter, indicating that NGFR inhibition promotes cell death (new **Extended Data Figure 5d, f**). We also sought to understand how NGFR signaling can cause resistance. Upon investigation of the ligands for NGFR (which are several neurotrophins), we observed that TR cells produce increased levels of Brain-Derived Neurotrophin Factor (BDNF; new **Figure 6e**). Importantly, we demonstrate that BDNF depletion increases T cell sensitivity in TR cells (new **Figure 6f** and **Extended Data Figure 5g**). These results demonstrate that a defining mechanistic feature of TR cells is the induction of NGFR, which is stimulated by BDNF to drive the T cell-resistant phenotype.

6) *In Figure 6b, does shNGFR already exhibit an effect in non-TR cells? This control experiment needs to be included.*

Author reply: We have now performed this experiment. Consistent with their commonly low NGFR expression, we did not see a T cell-sensitizing effect of shNGFR in parental cells (new **Extended Data Figure 5c**).

7) *Figure 5: what about NGFR expression in biopsies from resistant patients?*

Author reply: The patients represented in Figure 5 were not treated with immunotherapy, so we can unfortunately not assess this question in this cohort. To address this issue, we stained a second cohort of patients who were either sensitive or resistant to anti-PD1. NGFR-high tumors were significantly enriched for a lack of clinical benefit to anti-PD1 treatment (new **Extended Data Figure 4c**), consistent with our original observations.

8) *Minor point: in general, the labelling of figure axes could be improved; as an example, in Figure 3b, it's not clear immediately that NGFR MFI is measured, etc.*

Author reply: We thank the reviewer for pointing this out; we have improved the legend annotations now.

Reviewer #2 (Remarks to the Author); expert on PDX models, melanoma and immunotherapy:

This is a nice manuscript with some interesting findings.

Author reply: we thank the reviewer for his/her enthusiasm of the manuscript. In response to his/her comments, we have added several experiments to the manuscript; please see detailed replies below. In total, we have included 7 new Main Figure Panels and 11 new Extended Data Figure panels.

I have only three major concerns:

1. *The immunotherapy experiments are not conducted with autologous tumor and TILs. Although HLA matched it is therefore risky that some of the effects are not general but only pertaining to allogeneic responses or regulation of antigen processing. The authors should consider repeating some of the data using paired autologous TILs and tumor cells.*

Author reply: To answer the reviewer's question, we performed several experiments addressing these potential issues. First, we assessed whether allogeneic responses play a major role in our experimental settings. We observed that at the highest co-culture ratio of T cells : tumor cells (which is 1:1 and regularly used), there was no killing in the untransduced cells (**Reviewer Figure 1a**). In contrast, the MART-1 TCR-transduced T cells killed D10 cells effectively (but not TR cells, as expected, **Reviewer Figure 1a**). This result indicates that, at least at the ratios we normally use for our co-cultures, allogeneic responses of donor T cells do not play a major role.

Next, we treated D10 tumor cells with increasing concentrations of untransduced T cells up to 30 T cells per tumor cell, far exceeding our standard ratios. Even in this setting, there was no significant killing observed (**Reviewer Figure 1b**). Also this result indicates that the effects we see in our co-cultures are not attributed to allogeneic responses. Of note, we use untransduced T cells as a control arm in many experiments including the *in vivo* experiments, which should also account for any potential allogeneic responses.

Figure 1. (a) Colony formation assay of melanoma cells without T cells, with untransduced T cells (1:1 ratio with tumor cells) or with MART-1 TCR-transduced T cells from the same donor. Only MART-1 specific T cells were able to kill tumor cells. (b) Quantification of increasing ratios of untransduced T cells vs. tumor cells. Statistical analysis was performed using one-way ANOVA.

Regarding the second point, we fully agree with the reviewer that regulation of antigen processing is an important factor to consider. Therefore, we performed several experiments to exclude this issue. We performed experiments in settings where tumor cells rely on antigen processing in order to be recognized by T cells (e.g. **Extended Data Figure 1a**), but also settings where antigen processing is bypassed in the form of exogenous direct peptide loading (which loads the antigenic peptides directly onto HLA-A2 molecules on the tumor cell surface). Several key experiments were performed using both techniques (e.g. **Figure 6b**, **Extended Data Figure 5b**). The results demonstrate that cross-resistance to various antigen-specific T cells, as well as a T cell-sensitizing effect of NGFR knockdown is antigen processing-independent. Together, we feel that these combined approaches can convince us that antigen processing defects are not the cause of T cell resistance in the TR cells.

2. The extended data 5 suggest that there might be an off target problem with the shRNA targeting NGFR. Please repeat the experiment with CRISPR/Cas9 deletion of NGFR.

Author reply: We have now included also an experiment using CRISPR/Cas9 perturbation of the NGFR gene. Also this type of NGFR inactivation promotes T cell sensitivity in a caspase-dependent manner (new **Extended Data Figure 5e, f**). These results are in line with our shRNA data.

3. Although the data are suggestive of the NGFR is necessary, which can be strengthened by CRISPR, is NGFR sufficient to drive resistance? If NGFR is transgenically overexpressed, will this result in resistance?

Author reply: We thank the reviewer for this interesting idea. We performed this experiment, the result of which shows that ectopic expression of NGFR promotes T cell resistance of melanoma cells (new **Figure 6c, d**), in line with the role of NGFR as a resistance driver.

REVIEWER COMMENTS

Reviewer #1 (Remarks to the Author):

In the revised version of their manuscript, Boshuizen and colleagues have incorporated an extensive set of additional experiments, including CRISPR/Cas9-mediated NGFR KOs and ectopic NGFR expression, which all in all considerably strengthened their study. Together with some new mechanistic assays, the experiments nicely support their finding that NGFR confers T cell resistance to melanoma cells in vitro.

It would have even further increased the impact of the paper if the authors had addressed whether NGFR inhibition can prevent melanoma resistance formation in vivo or, vice versa, whether NGFR overexpression in parental melanoma cells is sufficient to drive resistance in vivo. In any case, I feel that the study is now acceptable for publication. However, I suggest to clarify, for instance in the Discussion, line 376, that "ectopic expression of NGFR conferred T cell resistance to parental cells IN VITRO" or: "...to parental cells, at least in cell culture".

Furthermore, I feel that the title is not very clear; the authors might want to consider rephrasing it.

Reviewer #2 (Remarks to the Author):

The authors did an excellent job in answering my comments. /Jonas Nilsson

Reviewer #3 (Remarks to the Author):

While I congratulate the authors for the present manuscript in which they use very complete in the array of techniques and analyses, the part of the study regarding the RNAseq experiment and analysis is quite lacking. While I understand that this manuscript is late in the review process, there are several issues regarding the RNAseq part that need to be addressed.

For reasons there is this common line of thinking that in all cases PCR duplicates need not be removed for RNAseq data, which is not true at all. It all depends on the scenario:

1) Amount of starting RNA: if there is enough starting RNA in your samples (~200ng or more), removing PCR duplicates does not have a big effect, and can even be detrimental depending on the read length. But if the starting material is scarce, not removing the PCR duplicates will incur into artificially inflated gene expression measures due to the low diversity of the starting library depending on the target number of reads sequenced.

2) Length of reads and Paired End (PE) vs Single End (SE): in SE, the ability to discern PCR duplicates from reads that just look like a PCR duplicate but is not is seriously hampered. Similarly, the shorter the reads, the more difficult to discern true PCR duplicates is. The authors have performed SE 65bp read sequencing which is one read and very short (common practice for RNAseq is 75x2 and optimal is 150x2). Since the authors are in the low end (and only SE) they need to have really good starting material quality and amount to justify not removing PCR duplicates.

3) Related to the last point, the length of read also critically affects multimapping rate of reads, which when high, then also artificially inflate the expression levels of certain genes with high homology between them.

Hence the authors need to provide a supplementary table with the RNA extraction metrics to discern whether it is ok to leave the PCR duplicates in or they need to be removed. This needs to be stated in the M&M section as a sentence, specifying the average amount of RNA extracted from samples and also specifying whether PCR duplicates need to be removed or not.

Furthermore, a supplementary table with STAR alignment metrics need to be provided. STAR outputs all these metrics as a normal part of its output. Critical metrics to show would be total raw reads, mapped reads, Total splices and break down stats for those splices, mismatch, deletion and

insertion rates, multimaps and multimaps percentages as well as percentage of high multimap reads, percentage of reads that are too short, and finally total reads which are PCR duplicates and the percentage of those (whether they are removed or not, as stated above).

Regarding the multimapping reads they also need to provide explanation as of they were handled (i.e. what threshold of percentage of multimaps was used to discard reads as "high multimap reads")

Lastly the authors mention that they have used the "default" settings of STAR. The default is single pass mode, which is far less accurate than the two pass mode of STAR. This needs to be specified in the M&M section, and if it is indeed the case that they have done single pass alignment with STAR, they need to provide an explanation as of why they have instead of using the two pass mode.

All this information is critical to the reproducibility of the study and the evaluation of the quality of the analysis done in the RNAseq part of it.

Reviewer #3 (Remarks to the Author):

While I congratulate the authors for the present manuscript in which they use very complete in the array of techniques and analyses, the part of the study regarding the RNAseq experiment and analysis is quite lacking. While I understand that this manuscript is late in the review process, there are several issues regarding the RNAseq part that need to be addressed.

Author reply: We thank the reviewer for their enthusiasm of the manuscript. We have now included additional information and specifications in the Material & Methods section to address the reviewer's concerns regarding the RNA sequencing quality.

For reasons there is this common line of thinking that in all cases PCR duplicates need not be removed for RNAseq data, which is not true at all. It all depends on the scenario: 1) Amount of starting RNA: if there is enough starting RNA in your samples (~200ng or more), removing PCR duplicates does not have a big effect, and can even be detrimental depending on the read length. But if the starting material is scarce, not removing the PCR duplicates will incur into artificially inflated gene expression measures due to the low diversity of the starting library depending on the target number of reads sequenced.

Author reply: We completely agree with the comments from the reviewer that removal of PCR duplicates is needed in cases of samples with low quality and/or low amounts of RNA. For all the parental and TR cells for which we generated RNA sequencing data, at least 5000ng RNA was isolated, which is abundantly sufficient for RNA sequencing analysis (see reviewer **Table 1** below). Furthermore, these samples had an RNA Integrity Number (RIN) of at least 9.8, indicating high-quality RNA. As such, in line with the reviewer's suggestion, removal of PCR duplicates was not deemed necessary.

Cell line	Parental or TR?	RIN	Amount (ng)
D10	Parental	9.8	>5000
D10	TR	10	>5000
SkMel-23	Parental	10	>5000
SkMel-23	TR	10	>5000
M009R.X1.CL	Parental	10	>5000
M009R.X1.CL	TR	10	>5000

Reviewer table 1: RIN and amount of RNA extracted from cell lines for RNAseq.

2) Length of reads and Paired End (PE) vs Single End (SE): in SE, the ability to discern PCR duplicates from reads that just look like a PCR duplicate but is not is seriously hampered. Similarly, the shorter the reads, the more difficult to discern true PCR duplicates is. The authors have performed SE 65bp read sequencing which is one read and very short (common practice for RNAseq is 75x2 and optimal is 150x2). Since the authors are in the low end (and only SE) they need to have really good starting material quality and amount to justify not removing PCR duplicates.

Author reply: As indicated above, the starting material (RNA) was indeed of high quality and abundant amount.

3) Related to the last point, the length of read also critically affects multimapping rate of reads, which when high, then also artificially inflate the expression levels of certain genes with high homology between them.

Hence the authors need to provide a supplementary table with the RNA extraction metrics to discern whether it is ok to leave the PCR duplicates in or they need to be removed. This needs to be stated in the M&M section as a sentence, specifying the average amount of RNA extracted from samples and also specifying whether PCR duplicates need to be removed or not.

Furthermore, a supplementary table with STAR alignment metrics need to be provided. STAR outputs all these metrics as a normal part of its output. Critical metrics to show would be total raw reads, mapped reads, Total splices and break down stats for those splices, mismatch, deletion and insertion rates, multimaps and multimaps percentages as well as percentage of high multimap reads, percentage of reads that are too short, and finally total reads which are PCR duplicates and the percentage of those (whether they are removed or not, as stated above).

Author reply: We have added this information in the methods section of the manuscript. Also, we provided a supplementary table as per suggestion of the reviewer (new **Supplementary Table 3**) with the RNA extraction metrics.

Regarding the multimapping reads they also need to provide explanation as of they were handled (i.e. what threshold of percentage of multimaps was used to discard reads as “high multimap reads”)

Author reply: For this the default settings of HTseq-count were used. This means that no multi-mapping genes were used to calculate the read count. Why this is chosen in HTseq-count is described in their documentation which we copied below. The accurate calculation of the ratio of a gene between multiple samples (in our case the parental and TR cell lines) is most important. Therefore, these settings best fit our analysis.

Copied text from the HTseq-count documentation:

htseq.readthedocs.io/en/release_0.11.1/count.html#frequently-asked-questions

“Why are multi-mapping reads and reads overlapping multiple features discarded rather than counted for each feature?”

The primary intended use case for htseq-count is differential expression analysis, where one compares the expression of the same gene across samples and not the expression of different genes within a sample. Now, consider two genes, which share a stretch of common sequence such that for a read mapping to this stretch, the aligner cannot decide which of the two genes the read originated from and hence reports a multiple alignment. If we discard all such reads, we undercount the total output of the genes, but the ratio of expression strength (the “fold change”) between samples or experimental condition will still be correct, because we discard the same fraction of reads in all samples. On the other hand, if we counted these reads for both genes, a subsequent differential-expression analysis might find false positives: Even if only one of the gene changes increases its expression in reaction to treatment, the additional read caused by this would be counted for both genes, giving the wrong appearance that both genes reacted to the treatment.”

Lastly the authors mention that they have used the “default” settings of STAR. The default is single pass mode, which is far less accurate than the two pass mode of STAR. This needs to be specified in the M&M section, and if it is indeed the case that they have done single pass alignment with STAR, they need to provide an explanation as of why they have instead of using the two pass mode.

Author reply: The reviewer is correct that the one-pass mode is seen as default. We were under the impression the two-pass mode was the default. For the RNA sequencing analysis in this paper the two-pass mode was used. We have changed this in the Material & Methods section.

All this information is critical to the reproducibility of the study and the evaluation of the quality of the analysis done in the RNAseq part of it.

Author reply: We thank the reviewer for his/her comments.

REVIEWERS' COMMENTS:

Reviewer #3 (Remarks to the Author):

The authors have satisfactorily answered all queries from this reviewer.